# A comprehensive Earth System Model (AWI-ESM2.1) with interactive icebergs: Effects on surface and deep ocean characteristics

Lars Ackermann[1], Thomas Rackow[2, 1], Kai Himstedt[3], Paul Gierz[1], Gregor Knorr[1], and Gerrit Lohmann[1, 4]

[1] Alfred-Wegener-Institut, Helmholtz-Zentrum für Polar- und Meeresforschung, Bremerhaven, Germany
[2] European Centre for Medium-Range Weather Forecasts, Bonn, Germany
[3] German Climate Computing Center, Hamburg, Germany
[4] University of Bremen, Bremen, Germany

**Correspondence:** Lars Ackermann (lars.ackermann@awi.de)

**Abstract.** The explicit representation of cryospheric components in Earth System models has become more and more important over the last years. However, there are only few advanced coupled Earth System models that employ interactive icebergs, and most iceberg model studies focus on iceberg trajectories or ocean surface conditions.

Here, we present multi-centennial simulations with a fully coupled Earth System model including interactive icebergs to assess the effects of heat and freshwater fluxes by iceberg melting on deep ocean characteristics. The icebergs are modeled as Lagrangian point particles and exchange heat and freshwater fluxes with the ocean. They are seeded in the Southern Ocean, following a realistic present-day size distribution. Total calving fluxes and the locations of discharge are derived from an ice sheet model output which allows for implementation in coupled climate-ice sheet models.

The simulations show a cooling of up to 0.2 K of deep ocean water masses in all ocean basins that propagates from the southern high latitudes northward. We also find enhanced deep water formation in the continental shelf area of the Ross Sea, a process commonly underestimated by current climate models. The vertical stratification is weakened by enhanced sea-ice formation and duration due to the cooling effect of iceberg melting, leading to a 10% reduction of the buoyancy frequency in the Ross Sea. The deep water formation in this region is increased by up to 10%. By assessing the effects of heat and freshwater fluxes individually, we find latent heat flux to be the main driver of these water mass changes. The altered freshwater distribution by freshwater fluxes and synergetic effects play only a minor role. Our results emphasize the importance of realistically representing both heat and freshwater fluxes in the high southern latitudes.

## 1 Introduction

Icebergs play a crucial role in Earth's climate system. Their calving from Greenland and the Antarctic continent contributes significantly to the mass balances of the two ice sheets. For Greenland, approximately 550 Gt year$^{-1}$, representing a third to half of its freshwater release, is due to discharge (Enderlin et al., 2018). For Antarctica, values for iceberg discharge range from over 2,000 Gt year$^{-1}$ (Jacobs et al., 1992) to more recent estimates of approximately 1,300 Gt year$^{-1}$ (Depoorter

et al., 2013). Icebergs transport large amounts of fresh water, alter ocean salinity and temperature, and hence affect deep-water and sea-ice formation (Grosfeld et al., 2001; Stern et al., 2015). In regions of iceberg melting, the freshwater release leads to a freshening of the upper ocean, increasing the oceans' freezing temperature and enhancing stratification. Another direct effect is the cooling of the upper ocean layers by sensible and latent heat fluxes, increasing oceans' density and thus potentially decreasing stratification of the water column, which could counteract the effect of added freshwater. Despite their importance, icebergs are rarely represented in Earth System Models (ESM) in detail, and if accounted for, their effects on ocean conditions are often only parameterized (Devilliers et al., 2021). Freshwater fluxes from iceberg melting are distributed either homogeneously over a specific area or are treated as surface runoff, entering the ocean directly at coastal regions. The drawbacks of both methods are 1) the neglection of ocean dynamical effects on the icebergs and hence an unrealistic spatial distribution of freshwater release, 2) missing sensible and latent heat feedback from icebergs to the ocean and vice versa, and 3) neglecting iceberg size-dependent dynamics and impacts on the northward extent of the freshwater release and the associated cooling by giant icebergs (Rackow et al., 2017).

Early studies using global ocean models with implemented Lagrangian iceberg models showed a good representation of iceberg trajectories (Bigg et al., 1997; Gladstone et al., 2001). Later studies included interactive icebergs with heat and freshwater feedback into fully coupled ESMs of varying complexity. Jongma et al. (2009) used an Earth System model of intermediate complexity (Claussen et al., 2002). In a simulation with interactive icebergs, they found a decrease in sea ice concentration and associated warming in the Weddell Sea, compared to a control run with freshwater homogeneously distributed over the Southern Ocean. Simulations using more advanced models with somewhat higher resolutions of $1°x1°$ for the ocean component were done e.g. by Martin and Adcroft (2010) and Stern et al. (2016). In comparison to a simulation with iceberg freshwater fluxes parameterized as surface runoff, Martin and Adcroft (2010) found a freshwater export via icebergs from coastal regions resulting in positive salinity anomalies and enhanced deep convection. They also found a decreased sea-ice cover. Using a more realistic size distribution, including larger km-scale icebergs, Stern et al. (2016) found a total decrease in sea ice concentration but cooling and freshening of the Weddell Sea. They argue to focus on large icebergs as these have the most significant effect on temperature and salinity changes. Rackow et al. (2017) add to this point by showing how the inclusion of even larger, giant icebergs impacts the meridional distribution of the iceberg meltwater input in their model simulations. Model simulations with even higher horizontal resolution of about $0.25°x0.25°$ were performed with ocean-only models (Marsh et al., 2015; Merino et al., 2016). They show the importance of icebergs for a realistic representation of Southern Ocean sea ice and its freshwater balance. However, heat fluxes from iceberg fusion were neglected. An overview of coupled climate-iceberg models is given in Tab. 1.

So far, most studies have focused on surface conditions in the Southern Ocean. However, the effect of interactive icebergs on deep ocean water masses' characteristics has received less attention due to the necessary long time scales and the associated high computational costs. This question seems especially important concerning the known deep ocean warm biases in models participating in the Coupled Model Intercomparison Project (CMIP) (Rackow et al., 2019), which could affect long-term future and paleoclimate simulations e.g. by their ability to store heat in the abyssal ocean. Warm deep-ocean biases are common

among complex Earth System models as found for FESOM by Sidorenko et al. (2019) and Streffing et al. (2022), and other climate models (e.g., Delworth et al., 2006, 2012; Jungclaus et al., 2013; Rackow et al., 2019; Sterl et al., 2012).

This study is the first to investigate the contribution of iceberg freshwater and heat fluxes on deep ocean properties in a complex Earth System model. We combine a fully coupled ESM with interactive icebergs in the Southern Ocean, using a resolution as high as $\sim 1/3°$ in coastal areas, with a size distribution representing present-day iceberg observations. We use the latest version of the Alfred Wegener Institute-Earth System Model (AWI-ESM) with an interactive Lagrangian iceberg model. While the model allows for interactive icebergs in both hemispheres, our simulations only include iceberg in the Southern Ocean. We acknowledge the potential implications of iceberg-related freshwater and heat fluxes for deep-water formation in the North Atlantic and, hence, on AMOC. However, our primary focus is an enhanced understanding of processes involved in climate-iceberg interactions rather than simulating realistic climatologies.

This study is organized as follows: section 2 describes new developments in the climate and iceberg model as well as the calving mechanism. Furthermore, the simulation setups are summarized. Section 3 analyses the model results from different simulations with respect to iceberg dynamics and the effects of heat fluxes and the differing freshwater flux distribution, as well as synergetic effects on deep ocean characteristics. We discuss our results in Sect. 4, and a conclusion is given in Sect. 5.

**Table 1.** Climate models with interactive iceberg component and freshwater feedback. *model* gives the model name. If ocean components differ between model versions, it is indicated by brackets; *hor. resolution* and *vert. levels* gives the horizontal resolution and number of vertical levels; *HF cpl.* states whether heat flux feedback from iceberg melting is included; *seeding* states the size-distribution used, for the Southern Hemisphere: GL - Gladstone et al. (2001), TO - Tournadre et al. (2016), and for the Northern Hemisphere: BI - Bigg et al. (1996); Dowdeswell et al. (1992); *run length* gives the integration time of the simulations; [1]) Jongma et al. (2009) used a 900 yr spinup, [2]) Marsh et al. (2015) state that heat flux feedback is implemented but turned off, [3]) UKESM uses NEMO-ICB as the ocean model and hence allows for heat flux feedback, however, no statement is given whether it is turned on or off, [*]) the model is of intermediate complexity (EMIC), [**]) total discharge is given by an interactive ice sheet model.

| model | hor. resolution / vert. levels | HF cpl. | seeding | run length [yr] | references |
|---|---|---|---|---|---|
| ECBilt-CLIO[*] | 3° / 20 | yes | GL | 100[1] | Jongma et al. (2009) |
| GFDL CM2G (GOLD) | 1° / 63 | yes | GL | 120 | Martin and Adcroft (2010) |
| GFDL CM2G (MOM6) | 1° / 63 | yes | TO | 120 | Stern et al. (2016) |
| iLOVECLIM[*] | 3° / 20 | yes | BI[**] | 12,000 | Bügelmayer et al. (2015) |
| LOVECLIM[*] | 3° / 20 | yes | uniformly[**] | future scenarios | Schloesser et al. (2019) |
| NEMO-ICB | 0.25° / 75 | no[2] | GL | 20 / 30 | Marsh et al. (2015); Merino et al. (2016) |
| UKESM | 1° / 75 | -[3] | GL[**] | 45 + future scenario | Siahaan et al. (2022); Smith et al. (2021) |

## 2 Methods and model description

The model used for this study is the AWI Earth System Model (AWI-ESM-2.1) with interactive icebergs. It consists of the AWI Climate Model (Rackow et al., 2018; Sidorenko et al., 2015), but comprises a newer version of the ocean model FESOM. It also uses dynamic vegetation (Reick et al., 2013). Its atmosphere component is the European Centre for Medium-Range Weather Forecasts' Model in Hamburg (ECHAM6) in its sixth generation (Stevens et al., 2013): A general circulation model run with the T63L47 setup, i.e., approximately a $1.9°$ horizontal resolution and 47 layers in the vertical.

ECHAM includes a hydrological discharge model for river runoff (Hagemann and Dümenil, 1997). On a subgrid-scale, surface runoff is transported along river routes and released to the ocean domain at the mouth of the river. Runoff is assumed to be liquid. Although a snow layer model is included, snow processes are not considered explicitly in areas that are impermeable to water infiltration, including glacial areas (Reick et al., 2021). Here, excess precipitation, including snowfall, is just added to the surface runoff as liquid freshwater. However, latent heat fluxes only include evaporation and sublimation according to the atmospheric water vapor. Hence, the river discharge implicitly accounts for the mass balance of glaciers, including freshwater fluxes from iceberg discharge and basal melting. But latent heat fluxes from iceberg melting are not accounted for. A detailed description of the land-surface and hydrological discharge components can be found in (Reick et al., 2021).

The ocean model used here is version 2 of FESOM, the Finite-VolumE Sea ice-Ocean Model (FESOM2). In contrast to its predecessor (Wang et al., 2014), it now employs the finite volume method instead of finite elements which allows for higher computational efficiency (Danilov et al., 2017). The model uses unstructured meshes that enable efficient high-resolution modeling of highly dynamic regions while leaving a coarser resolution in other regions. The mesh used in this study shows a horizontal resolution of up to 20 km in the high latitudinal coastal regions and a coarser resolution of around 120 km in the low latitudes. It has been widely used (e.g., Danabasoglu et al., 2016; Sein et al., 2016; Wang et al., 2016a, b) and is applicable for long-term simulations. The iceberg component runs as a submodel of the ocean–sea-ice model FESOM2 (Danilov et al., 2017; Koldunov et al., 2019; Scholz et al., 2019, 2022). In contrast to a previous version of this model introduced by Rackow et al. (2017), freshwater and heat fluxes are now interactive, providing a new level of coupled feedbacks. The initial position, number, and proportions of icebergs are derived from an ice-sheet model (ISM) output, allowing future applications in a coupled climate-ice sheet setup. The iceberg size distribution follows a $-3/2$ power law, derived from satellite observations for both open-ocean and near-coastal areas (Barbat et al., 2019; Tournadre et al., 2016).

### 2.1 The iceberg module

The iceberg component is a submodel of FESOM. However, bi-directional coupling between the ocean and the iceberg had yet to be implemented in the model. Studies using the interactive iceberg component were ocean-only simulations, in which the icebergs were treated as passive tracers that allowed diagnosing a meltwater field (Rackow et al., 2017), but lacking freshwater and heat feedback to the ocean model. This work introduces the iceberg module as a fully coupled component within FESOM. Hence, freshwater and heat fluxes are bi-directionally coupled between icebergs and the ocean.

Initial iceberg positions and dimensions are obtained via fields of calving discharge from the ice sheet (see section 2.2 for details). The model is a Lagrangian iceberg model, i.e., all icebergs are represented by point particles. While these particles are zero-dimensional, each has a length, width, and height assigned to it. These physical quantities are altered during the simulation by thermodynamical processes. For simplification, each iceberg is assumed to have a quadratic base area and to be of cuboidal shape. Thermodynamics take into account the erosion by surface waves and buoyant convection mainly following work by Bigg et al. (1997), Gladstone et al. (2001), and Martin and Adcroft (2010), as well as basal and lateral "basal" melting following the three-equation formulation by Hellmer and Olbers (1989) and Holland and Jenkins (1999). Lateral "basal" melting here refers to melting on the submerged sides of an iceberg by turbulent heat transfer analogously to the melting at the base of an iceberg. It is not to be confused with buoyant convection. The implementation is motivated by Bigg et al. (1997) and is described in Rackow et al. (2017). The wave erosion follows an empirical formulation, taking into account the sea surface temperature, the relative velocity between wind and ocean, and a damping factor that is a function of sea ice cover. The buoyant convection also follows an empirical formulation and depends on the "thermal driving" temperature $T_d = \max(0, T_m - T_f)$, with $T_f$ being the in situ freezing temperature at mid-depth, and $T_m$ being the average water temperature along the iceberg draft. The basal (and lateral "basal") melt rates are derived by solving the energy and salinity balances in the boundary layer at the iceberg-ocean interface given far-field- temperature and salinity values. For the basal melting, temperature and salinity at iceberg depth are taken, and for the lateral "basal" melting, temperature and salinity are averaged along the iceberg depth. This approach allows for negative melt rates, i.e., freezing at the iceberg base. Besides accounting for the wind drag, no exchange processes with the atmosphere are modeled, particularly no surface mass balance and radiative or conductive heat fluxes. The surface melt due to radiation is of minor importance compared to oceanic-driven melt rates (Bigg et al., 1997). A detailed description of the model can be found in Rackow (2011) and Rackow et al. (2017). Interactions between icebergs are not modeled but are parameterized in a very simple manner to avoid an over-loading of ocean cells: If an iceberg is about to change from one grid element to another, the total iceberg area contained in this grid element is summed up. If the new iceberg leads to a larger total iceberg area than the actual element area, it does not change the grid element but stays in its previous grid element and is set back to its previous position. It can still move within the grid cell element or to a neighboring grid cell that is not saturated yet. Furthermore, model icebergs are not discharged into saturated ocean grid cells but are distributed over the coastal and neighboring grid cells within the respective basin (Fig. 1). Whenever the model iceberg's depth reaches deeper than the local bathymetry, it is assumed to be grounded, and its velocity is set to zero. Basal melting can still occur and eventually set the model iceberg free again once its depth is sufficiently reduced.

Different measures have been taken to speed up the iceberg module: The first is by implementing a "scaling approach" similar to Martin and Adcroft (2010). This approach reduces the number of simulated icebergs by dividing the icebergs into different size classes. For each class, a scaling factor is defined by which the number of simulated icebergs is reduced. Each simulated iceberg then represents multiple other icebergs. The calculated freshwater and heat fluxes are multiplied by the scaling factor to ensure mass and energy conservation (Appendix A). A second approach for speeding up the iceberg module is a variable coupling frequency between ocean and iceberg components. Initially, the coupling and, hence, the simulation of icebergs took

place every FESOM ocean time step. Due to the relatively slow movements of the icebergs, a coupling three or four times a simulated day seems to be sufficient instead of the one-to-one coupling that has been implemented previously.

Freshwater and heat fluxes from iceberg melting are added to the respective FESOM internal sea-ice fluxes. Hence, the iceberg feedback is applied to the ocean surface. Furthermore, it is distributed to all nodes that constitute the containing element. The calving discharge is compensated by a reduction of Antarctic surface runoff. As the calving flux is considered constant in our simulation setup, the surface runoff reduction is also considered constant and is done at every coupling time step between the atmosphere (land surface) and the ocean. The total salinity is held constant in FESOM internally, and local surface freshwater fluxes (like those from iceberg melting in our model setup) are balanced by a freshwater compensation distributed homogeneously over the whole ocean domain. Hence, both fluxes together, the iceberg melting and the reduced surface runoff, lead to a redistribution of freshwater from the coastline to the open ocean. While there is temporal variability in the iceberg melt fluxes, the compensating reduction of Antarctica's surface runoff is fixed over time. To account for this discrepancy and to ensure a consistent freshwater budget, the total salinity is balanced, so the iceberg setup has no additional net freshwater flux compared to the model version without interactive icebergs. However, the freshwater fluxes from iceberg melting are not considered part of the Antarctic surface runoff anymore, i.e., the influx is allowed to occur on the open ocean instead of directly along the coast and shelf regions. While the total salinity is balanced, oceans' total internal energy is not, and there is a negative net heat flux due to iceberg melting that is not accounted for in the model setup without interactive icebergs. Hence, a new climatological equilibrium is expected to develop compared to the default model setup without interactive icebergs.

## 2.2 Iceberg seeding and experimental setup

The initial conditions of each iceberg need to be provided, including the location, velocity, dimensions, and scaling factors. Apart from the velocities, which are set to zero initially, all other parameters are deduced from the ice sheet model output. For our study, this is the Parallel Ice Sheet Model (PISM) (Martin et al., 2011; Winkelmann et al., 2011). The model output provides a spatially continuous calving and discharge field on a 16x16 km grid (Fig. 1) with the highest calving rates per grid cell of up to 10 Gt year$^{-1}$ (approximately 45 m year$^{-1}$) along the Filchner-Ronne and the Ross ice shelves as well as in the Amundsen Sea, which corresponds well with observations (Depoorter et al., 2013).

To generate discrete icebergs from the continuous field, the calving flux is summed up over each basin (Fig. 1a) to get the integrated total amount of ice discharge within each basin. Next, this amount is divided by a reference iceberg height of 250 m to get a total calving area. As reference iceberg size to derive the total number of seeded icebergs, the median of the power-law distribution is used. Following Tournadre et al. (2016), individual iceberg areas are drawn from a power-law distribution. The initial size distribution is shown in Fig. 1b, with the vast majority of icebergs being rather small (0.01-1 km$^2$) and only a few being larger than 100 km$^2$. A maximum iceberg area of 400 km$^2$ is assumed to avoid iceberg areas being larger than an ocean grid cell. Model icebergs are assumed to have a quadratic surface. The iceberg height is set to be equal to the length and width, respectively, but not larger than 250 m. This new feature compared to the previous iceberg model version (in which iceberg height was set to 250 m) is implemented to reduce the risk of instantaneous grounding of newly seeded icebergs in shallow

water regions. In an iterative process, the dimensions are adjusted so that the total iceberg volume matches the integrated
discharge. An overview of this scheme is given in Fig. A1.

For each size class (Fig. 1b), a specific scaling value is set by which the number of icebergs in this size class is reduced
(Tab. A). The heat and freshwater fluxes released by this iceberg are then scaled up again accordingly. For the seeding, a list
of ocean grid cells is generated. For each ice sheet model grid cell in which calving occurs, the nearest ocean grid cell and
the neighbouring cells are added to this list. Direct coastal grid cells are then removed from this list to reduce the risk of
instantaneous grounding. So, model icebergs are spread out near the coast but are not seeded directly at the coast. For each
basin, model icebergs are then distributed over the ocean grid cells contained in the list. These steps are done for each basin
individually to ensure a consistent distribution along the coastline. While the same size-distribution is applied for each basin,
the actual model icebergs are drawn randomly from this particular distribution. Hence, the actual size-distribution may vary
for each grid point. The total calving flux of roughly 1,731 Gt year$^{-1}$ is subtracted from the surface runoff to ensure a closed
water balance (Fig. 1c).

To spin up the iceberg model, an equilibrated pre-industrial run has been continued with icebergs but with freshwater and
heat feedback turned off ($ICB_{spinup}$). This spinup was run for 100 years, after which the total iceberg melt flux balances the
calving flux (Fig. 1c). Several experiments were branched off from this spinup, a fully coupled run with icebergs (ICB), two
partially coupled iceberg runs, one without latent heat fluxes from iceberg fusion ($ICB_{FW}$), and one without iceberg meltwater
feedback ($ICB_{HF}$), respectively. The same iceberg setup has been used for these runs. Additionally, a control run without
icebergs (CTL) has been run. All runs are summarized in Table 2.

**Table 2.** Experiments run within the scope of this study. *EXP. ID* indicates the name used for this experiment throughout this study; *FW*
and *HF* indicate how freshwater and heat fluxes from iceberg melting are treated within the simulation; *cpl. frequency* indicates the coupling
frequency in FESOM time steps per iceberg submodel time step; *run length* indicates the length of the simulation in model years.

| EXP. ID | FW | HF | scaling | cpl. frequency [-] | run length [yr] |
|---|---|---|---|---|---|
| $ICB_{spinup}$ | surface runoff | absent | yes | 8 | 100 |
| CTL | surface runoff | absent | - | - | 700 |
| ICB | interactive | interactive | yes | 8 | 700 |
| $ICB_{FW}$ | interactive | absent | yes | 8 | 700 |
| $ICB_{HF}$ | surface runoff | interactive | yes | 8 | 700 |

## 3   Results

This section presents the results of a pre-industrial run with interactive icebergs as well as only partially coupled runs with
either freshwater or heat flux feedback, each running for 700 years. The results presented are averaged over the last hundred
model years of the simulations. Temperature and salinity fields of CTL with respect to the Polar Science Center Hydrographic
Climatology (PHC3.0) (Steele et al., 2001) are shown in Fig. 2. Strong warm biases in the deep Southern Ocean of up to

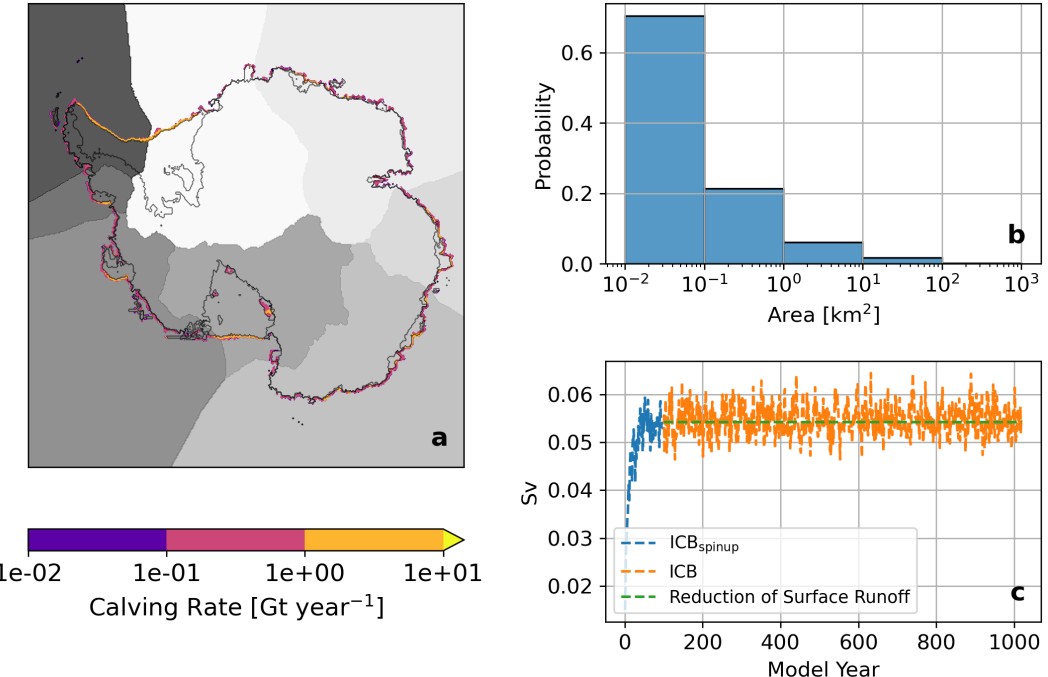

**Figure 1.** a) Calving flux from a PISM standalone simulation. The grey-shaded areas depict different basins for which the integrated total discharge is calculated individually; b) size distribution of seeded icebergs; c) iceberg-related freshwater flux for spinup and ICB, and the reduction of Antarctica's surface runoff.

$1-2$ K are present in CTL, most pronounced in the Atlantic and Indian Ocean sectors (Fig. 2a-c). Furthermore, there is a pronounced fresh bias in the continental shelf regions around Antarctica of up to $0.5$ psu (Fig. 2d-f). Deep ocean conditions of CTL and ICB with respect to (PHC3.0) (Steele et al., 2001) are shown in Fig. A2 and A3.

### 3.1 Trajectories

Figure 3a illustrates iceberg trajectories for the fully coupled iceberg run ICB. Two main pathways can be recognized: One branching off the Antarctic Peninsula, where small and large icebergs follow the Antarctic Circumpolar Current (ACC), and one pathway in the Ross Sea with medium-sized icebergs. A third branch of icebergs, escaping the ACC near the Kerguelen Plateau as recognizable in the observational data (Fig. 3d) and found by Rackow et al. (2017), is not present in our model results. Large icebergs tend to stay along the coast, following the Antarctic Coastal Current. The general patterns resemble satellite observations for giant icebergs by Budge and Long (2018) and Stuart and Long (2011) (Fig. 3d). However, model icebergs travel further north compared to observations. In the Ross Sea, their pathways are confined by the Antarctic Convergence Zone (indicated as the zone between the $2°C$ and $5°C$ SST isotherms). The spatial patterns of freshwater and heat fluxes (Fig. 3b

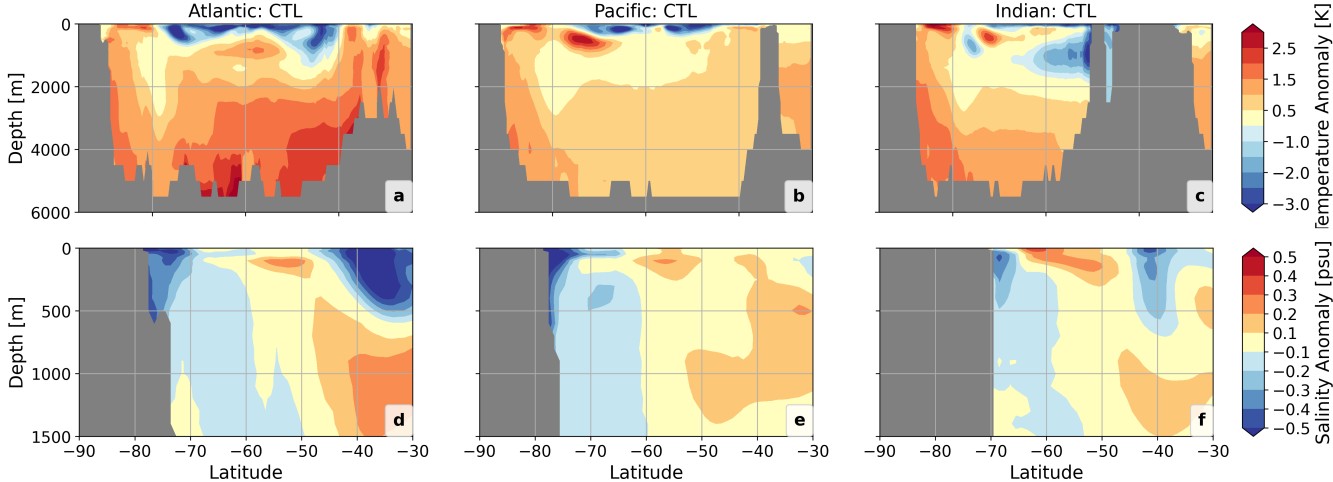

**Figure 2.** a-c) Zonal mean temperature anomaly for the Atlantic, Pacific, and Indian Ocean sectors, respectively, of CTL with respect to PHC3.0 (Steele et al., 2001); d-f) like a-c) but for salinity and limited to the upper 1,500 m and 30-90°S.

and c) match the trajectories and show melting hot spots near the coast, inside the Weddell Sea, and at the tip of the Antarctic peninsula where, very locally, freshwater and heat fluxes of over $10\,\mathrm{m\,year^{-1}}$ and $10\,\mathrm{Wm^{-2}}$, respectively, are reached.

### 3.2 Surface conditions

The anomalies for sea surface salinity (SSS), sea surface temperature (SST), and sea ice height are shown in Fig. 4 for ICB, $\mathrm{ICB_{HF}}$, $\mathrm{ICB_{FW}}$, and CTL with respect to the spinup. ICB, $\mathrm{ICB_{HF}}$, and $\mathrm{ICB_{FW}}$ show pronounced positive salinity anomalies in the shelf regions of the Weddell Sea (Fig. 4a and g). A similar salinity anomaly is detected in the Ross Sea sector in $\mathrm{ICB_{HF}}$. However, the underlying dynamics are fundamentally different. In ICB and $\mathrm{ICB_{FW}}$, the surface runoff is most reduced compared to CTL in areas that correspond to the coastal regions with the highest calving rates. In these areas, freshwater by iceberg calving is parameterized via the river routing scheme and eventually treated as river discharge in CTL and $\mathrm{ICB_{HF}}$. As the icebergs do not melt entirely in their regions of origin but rather further north off the coast, the model experiences a relative freshwater export from near-coast shelf regions, which results in pronounced positive salinity anomalies in the shelf regions of the Weddell Sea in ICB and $\mathrm{ICB_{FW}}$. In contrast, in $\mathrm{ICB_{HF}}$ (in which the surface runoff is not altered compared to CTL), enhanced sea ice formation (Fig. 4i) leads to increased brine rejection. This can be recognized in the Weddell Sea shelf region and the Ross Sea. These regions of positive salinity anomaly match well to the pattern of increased sea ice height for $\mathrm{ICB_{HF}}$ (Fig. 4g and I). Increased sea-ice cover in ICB and $\mathrm{ICB_{HF}}$ is very pronounced at the end of the summer season on the shelf regions (Fig. A4) and along the sea-ice edge at the end of the winter season (Fig. A5). In contrast, no systematic increase in sea ice height can be recognized in this region in ICB and $\mathrm{ICB_{FW}}$ (Fig. 4c and f). Here, the increased salinity due to reduced near-coastal freshwater surface runoff inhibits additional sea ice growth. But sea ice growth is fostered in coastal regions of the Amundsen and Bellinghausen Seas, along the Antarctic Peninsula, and along the Wilkes Land coast (Fig. 4c).

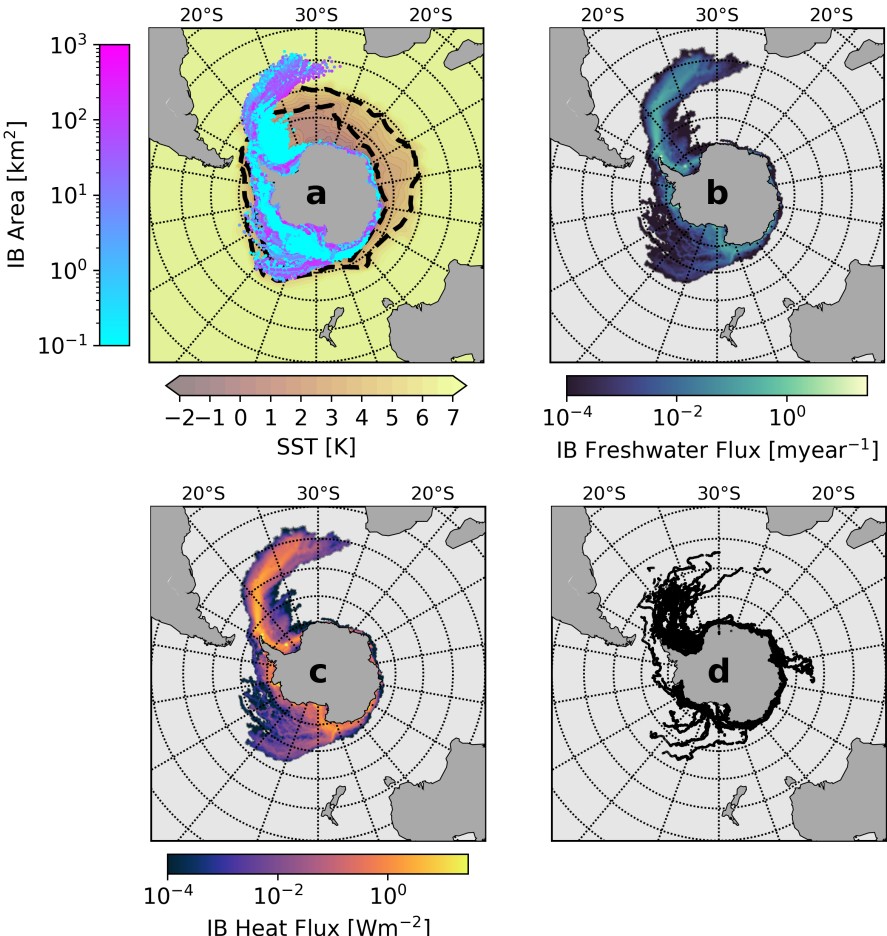

**Figure 3.** a) Sea Surface Temperature (SST) overlaid by iceberg trajectories with iceberg surface area shown on a logarithmic colorbar. The black dashed contour lines indicate the Antarctic Convergence Zone where SST falls from $5°C$ to $2°C$; b) freshwater flux due to iceberg melting; c) heat flux due to iceberg melting; d) satellite observations from the QuikSCAT portion of the Antarctic Iceberg Tracking Database (Budge and Long, 2018; Stuart and Long, 2011) over the period from 1991 to 2022. All model results are averaged over model years 600-700.

Here, freshwater and heat fluxes from iceberg melting are very high (Fig. 3b and c). Cooling patterns can be seen in the Weddell Sea and the Indian sector of the Southern Ocean (Fig. 4b and h), while warming is detected in the Amundsen and Bellinghausen Seas, as well as in the Ross Sea, leading to a dipole of warm/cold anomalies across the Antarctic Peninsula. The warming in the Amundsen and Bellinghausen Seas is linked to an increase in surface salinity that leads to enhanced vertical mixing and upward mixing of heat. In contrast to the similar cooling patterns in ICB and $ICB_{HF}$, a warming in the Weddell Sea can be recognized in $ICB_{FW}$ (Fig. 4e). In this experiment, an increase in salinity also leads to enhanced vertical mixing and convective mixing of heat as in ICB and $ICB_{HF}$, but latent cooling from iceberg melting is missing to compensate for this

230  surface warming. In general, the resulting responses for SST, SSS as well as sea ice height are dominated by the individual effects of heat and freshwater fluxes in ICB, revealing minor importance of synergetic effects on long time scales.

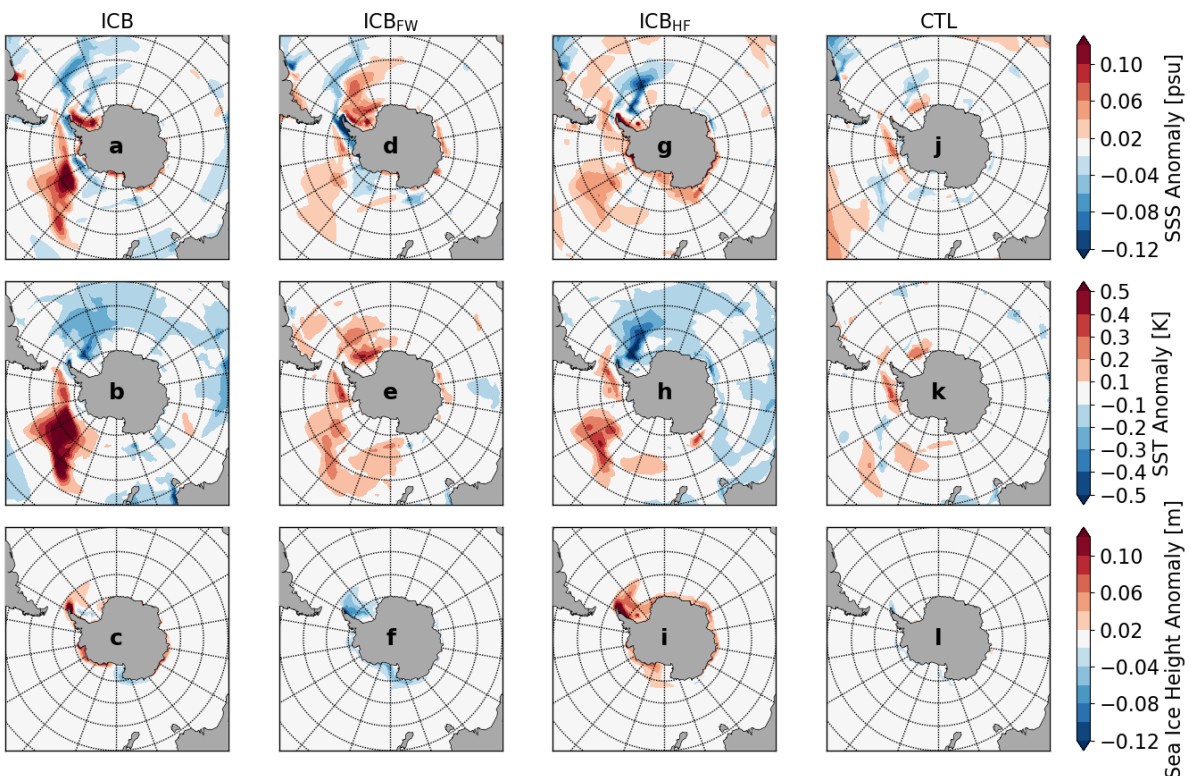

**Figure 4.** Multi-year anomalies of SSS, SST, and sea ice height for the experiments ICB (a-c), ICB$_{FW}$ (d-f), ICB$_{HF}$ (g-i), and CTL (j-l). All results are averaged over model years 600-700 and anomalies are calculated with respect to the spinup.

To assess atmospheric feedbacks, the mean zonal wind anomaly and the contributions to Antarctica's mass balance are shown in Fig. A6 and A7, respectively. A weakening of the westerlies south of 40°S of less than 10% can be seen in ICB compared to CTL. Regarding Antarctica's mass balance, the discharge (freshwater runoff via the river routing scheme) is reduced by
235  1,731 Gt year$^{-1}$ in ICB and ICB$_{FW}$ as mentioned in sec. 2.1 (Fig A7a). Changs in P − E and glacial melt are insignificant (Fig A7b and c).

### 3.3  Deep ocean conditions

Changes in deep ocean temperature for the Atlantic, Pacific, and Indian Ocean basins are illustrated in Fig. 5. After 700 model years, a cooling in all three basins can be seen for ICB and ICB$_{HF}$ with respect to the spinup run. The cooling of up to -0.2 K

is most pronounced in the Pacific (Fig. 5b and h). While no cooling is recognizable in $\text{ICB}_{\text{FW}}$, the patterns of ICB and $\text{ICB}_{\text{HF}}$ look very similar. The cooling signal extends from the surface layers of the Southern Ocean's Atlantic section (Fig. 5a and g) to the deep southern mid-latitudes. A cold cell can be seen in the North Atlantic at around 1,000 m depth. The deep North Atlantic, as well as the Arctic Ocean, show a warming trend. However, this is partly due to a general background trend and internal model variability as it is also visible for the control run (Fig. 5j). In contrast to the Atlantic basin, the cooling in the Pacific and Indian Oceans extends over the whole basins (Fig. 5b, c, h and i). Most pronounced in the Southern Ocean, it spreads more northward with depth. However, the upper ocean layers show a warming in the high latitudinal Pacific section of the Southern Ocean, corresponding to the warming of the Ross Sea (Fig. 4a, d, g). This effect is also visible in $\text{ICB}_{\text{FW}}$. Also here, CTL shows a slight warming (Fig. 5k), but to a much smaller magnitude than the other simulations.

A strong increase in salinity can be seen in ICB and $\text{ICB}_{\text{HF}}$ in the Atlantic and Pacific sectors from the surface to a depth of around 500 m (Fig. 6a,b and g,h). These salinity anomalies are mainly detected in the shelf regions of the Weddell and Ross Seas, indicating a link to the surface conditions (SSS anomalies in Fig. 4). However, $\text{ICB}_{\text{FW}}$ also shows positive SSS anomalies, especially in the Weddell Sea, but the vertical extension is limited to mixed layer depths (Fig. 6d). The main driver for the positive salinity anomalies reaching deeper levels is therefore attributed to the latent heat flux from iceberg melting. The effect of altered spatial freshwater distribution, on the other hand, plays a minor role.

## 3.4 Impact of HF and FW on adjustment time-scales

The effect of temperature changes on seawater density is small compared to the effects of salinity in our experiments. The salinity increase leads to a positive density anomaly and, hence, to a weakening of vertical stratification. This weakening is especially pronounced over the continental shelf in the Ross Sea in ICB and $\text{ICB}_{\text{HF}}$, and additionally along the coast of Wilkes Land in $\text{ICB}_{\text{HF}}$ (Fig. 7). $\text{ICB}_{\text{FW}}$ shows a strengthening of stratification around Antarctica except for the Weddell Sea. The change in the buoyancy frequency affects the magnitude of vertical mixing (Fig. A9) that is reduced (enhanced) in the Weddell Sea for $\text{ICB}_{\text{HF}}$ and ICB (CTL and $\text{ICB}_{\text{FW}}$). The increased vertical mixing in the open ocean part of the Amundsen Seas in ICB leads to an upward heat transport that results in surface warming (Fig. 4a).

This weakened vertical stratification in the Southern Ocean results in enhanced deep convection over continental shelves and enhanced formation of Antarctic Bottom Water (AABW) (Fig. 8). The AABW in the Indo-Pacific sector (AABW-IP) is increased by up to 10% in ICB and $\text{ICB}_{\text{HF}}$, though this change lies within one standard deviation. The AABW-IP strengthening occurs within the first 200 model years and stays at a rather constant level afterward. $\text{ICB}_{\text{FW}}$ and CTL show a weak trend. All simulations show a pronounced centennial variability in the AABW, indicating that this feature is due to internal model variability. As changes in AABW formation in the Atlantic sector only represent a minor contribution, global AABW mainly follows the AABW-IP signature originating in the Ross Sea shelf-region as the main area, which is affected by destabilized stratification due to iceberg heat fluxes. The global ocean temperature decreases by approximately 0.01 K per century over the first 400 years in ICB and $\text{ICB}_{\text{HF}}$. This corresponds with a rough calculation considering the enthalpy of fusion for a discharge flux of around 1, 700 Gt per year (Appendix). After 400 years, the cooling trend in ICB flattens while it continues to decrease in $\text{ICB}_{\text{HF}}$. The altered freshwater distribution via iceberg transport hence buffers the cooling. No cooling trend is recognizable

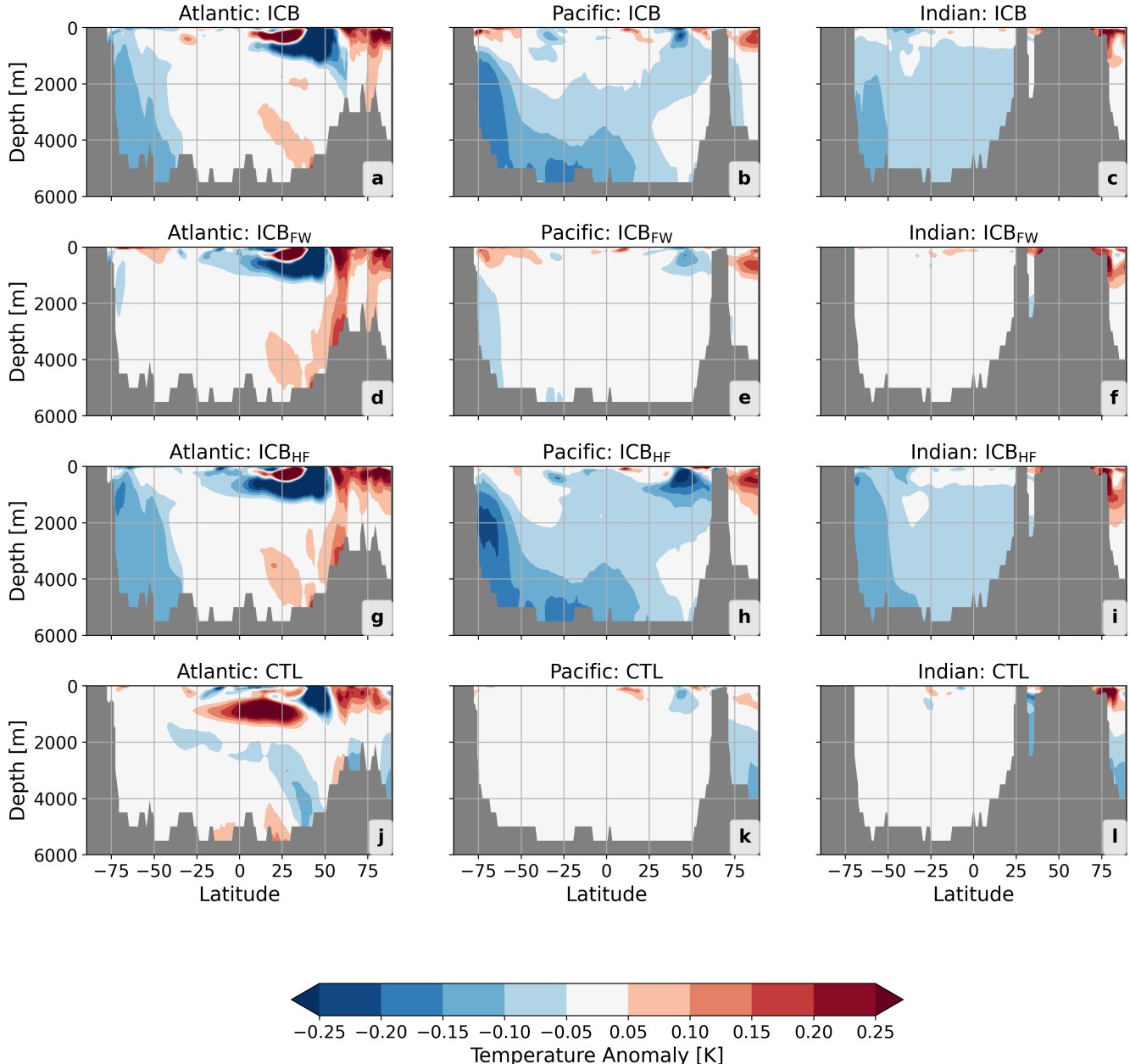

**Figure 5.** a-c: Temperature anomalies for the Atlantic, Pacific, and Indian Ocean, respectively for ICB; d-f: Temperature anomalies for $ICB_{FW}$; g-i: Temperature anomalies for $ICB_{HF}$; j-l: Temperature anomalies for CTL. All results are averaged over model years 600-700 and anomalies are calculated with respect to the spinup.

in CTL and $ICB_{FW}$. The Drake Passage throughflow is around 87-90 Sv in CTL, which is significantly smaller than suggested by observations (Donohue et al., 2016; Whitworth and Peterson, 1985). In our simulations, the Drake Passage throughflow

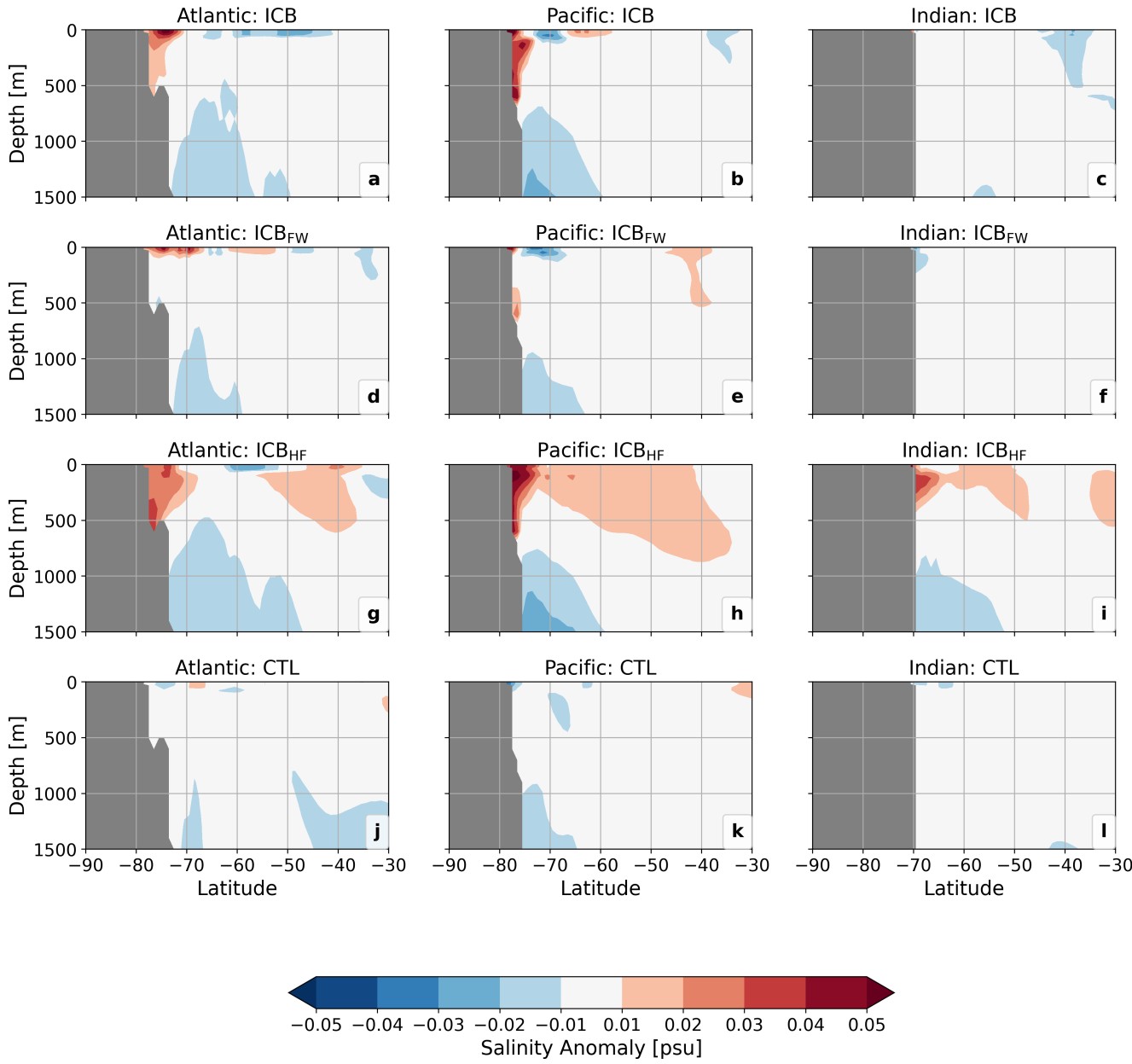

**Figure 6.** a-c: Salinity anomalies for the upper 1,500 m of the Atlantic, Pacific, and Indian Ocean sections of the Southern Ocean for ICB; d-f: Salinity anomalies for $ICB_{FW}$; g-i: Salinity anomalies for $ICB_{HF}$; j-l: Salinity anomalies for CTL. All results are averaged over model years 600-700 and anomalies are calculated with respect to the spinup.

increases to around 92 Sv in ICB. However, this increase lies within one standard deviation. The Weddell Gyre strength shows

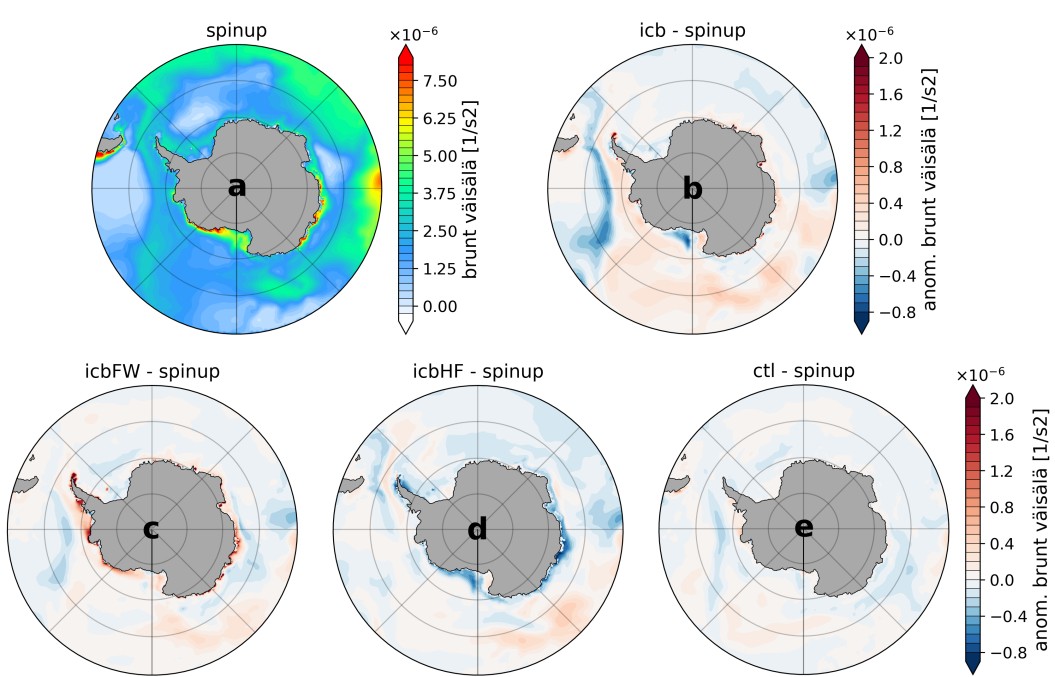

**Figure 7.** Brunt Väisälä frequency for spinup (a) and anomalies for ICB (b), ICB$_{HF}$ (c), ICB$_{FW}$ (d), and CTL (e) with respect to spinup averaged over the upper 250 m and for the model years 600-700.

similar variability and amplitude (80-89 Sv) for all simulations, and no clear differences between ICB and CTL can be seen. For comparison, the barotropic stream function is shown in Fig. A8.

## 4 Discussion

280 We have run multi-centennial simulations with a complex fully-coupled Earth System model, including interactive icebergs. While the iceberg trajectories show generally good agreement with observations, there are also some discrepancies. There are no icebergs branching off near the Kerguelen Plateau in our simulations as seen in observations (Fig. 3) or as found by Rackow et al. (2017) in their ocean-only simulations with prescribed atmospheric forcing. This might be due to the coarse resolution of the atmosphere and land surface. Steep orographic gradients are smoothed out, which hinders the formation of katabatic 285 winds. Instead, icebergs are mainly affected by polar easterlies and hence pushed onshore due to Ekman dynamics. Another reason could be that the threshold values for the "sea ice capturing mechanism" might need to be lowered/tuned after the switch from FESOM1 to FESOM2. In the earlier version, a sea ice strength of at least $P_s = 10,000\,\mathrm{Nm}^{-1}$ was assumed to allow this mechanism, and modeled sea ice strength in this area might be smaller in FESOM2. Rackow et al. (2017) had noted that the two largest giant icebergs in their simulation left the coastal current near the Kerguelen Plateau only because of being captured

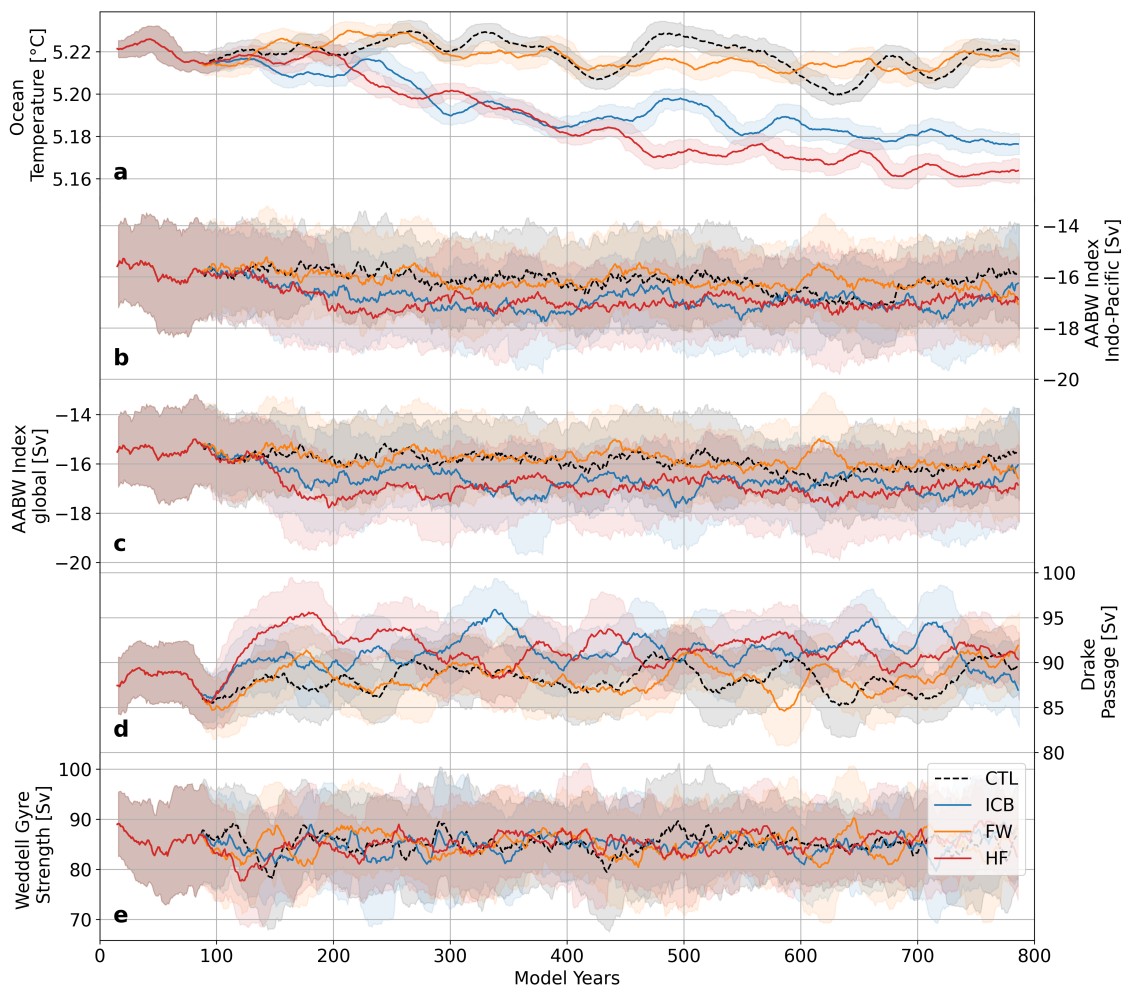

**Figure 8.** 50-year rolling mean timeseries for global ocean mean temperature (a), Antarctic Bottom Water in the Indo-Pacific basin (b) and globally (c) as the maximum stream function value at $30°$S, Drake Passage throughflow (d), and Weddell Gyre strength (e), defined as the difference between the maximum and the value at of the smallest closed contour line of the barotropic stream function, for ICB, $ICB_{HF}$, $ICB_{FW}$, and CTL; shaded areas indicate one standard deviation.

290 by the expanding sea ice, and thus being able to cross the southern ACC front (Orsi et al., 1995), which would be difficult just via the other model dynamics due to the tendency of giant icebergs to follow isolines of SSH. In general, large model icebergs

tend to be too confined to coastal regions. This was already found by Rackow et al. (2017), and other studies (e.g., Merino et al., 2016). When being able to leave the coast, the icebergs show a drift further north in our simulations than in observations but do not travel as long distances as in Rackow et al. (2017). However, the observational data used here only covers icebergs larger than $\sim 5 - 6$ km (Stuart and Long, 2011) and hence can miss substantial parts of the end of giant icebergs' trajectories. The iceberg model, on the other hand, does not include a breakup parametrization for large icebergs. Hence, the occurrence and longevity of large icebergs might be overestimated when compared to observations. As the dynamics of small and large icebergs differ (Rackow et al., 2017), a break-up parametrization would affect trajectories and melt patterns (England et al., 2020). The simple parametrization implemented in our model to avoid an overfilling of ocean grid cells leads to very long residence times of icebergs. Icebergs tend to accumulate in certain places, e.g. the tip of the Antarctic Peninsula (Fig. 3). They may block the pathway for more downstream icebergs when a grid cell is saturated, although other model icebergs are not taken into account in one single iceberg's momentum balance. The long residence times delay the escape to open-ocean waters where a breakup parametrization, like the "footloose" mechanism used in England et al. (2020), would come into play. In this way, large icebergs decay predominantly near-coast, and long trajectories, as mentioned in Rackow et al. (2017), are avoided.

Physical feedbacks besides freshwater and heat fluxes. These feedbacks may include effects on surface albedo, surface wind stress, and sea surface height. Furthermore, we used a uniform calving size distribution for all ocean basins. However, size distributions vary at different locations, and giant icebergs calve very rarely (Qi et al., 2021). Hence, they should be treated as statistically rare events similar to volcanic eruptions, for example, by calving them stochastically in ensemble simulations or by prescribing their time-mean effects via pre-computed melt climatologies (Stern et al., 2016; Rackow et al., 2017).

The effects on sea surface conditions support the findings of previous studies. Martin and Adcroft (2010) and Stern et al. (2016) also found warming in the Amundsen and Bellingshausen Seas as well as in the Ross Sea. This warming is explained by increased upward heat transport due to a destabilization of the upper ocean layer's stratification. This weakened stratification stems from increased salinity due to northward freshwater export. The warming in the open ocean part of the Amundsen Sea (Fig. 4b) is consistent with findings by Martin and Adcroft (2010). In our simulations, this warming is most pronounced in the upper $100$ m. The timeseries of this warm anomaly averaged between $110°W - 130°W$ and $55°S - 65°S$ shows a strong multi-centennial variability (Fig. A7d) and a consistent warming in ICB compared to CTL. However, the mechanism behind this pattern may be more complex as the warming occurs far off the coast and off iceberg trajectories (Fig. 3a), and no significant correlations with the Weddell Gyre strength (r $= -0.20$), the Drake Passage throughflow (r $= -0.12$), or the Southern Annular Mode (SAM) (r $= -0.20$) (Fig. A7e) are found.

In our ICB and $ICB_{FW}$ experiments, the surface runoff is reduced by the amount of iceberg discharge. This leads to positive salinity anomalies in the Weddell Sea (especially pronounced in the Weddell Sea shelf region) and the Ross Sea shelf region (Fig. 4a and d). However, our simulation $ICB_{HF}$ also shows a strong increase in salinity despite unaltered surface runoff. Hence, increased sea-ice formation and duration also play an important role in Ross Sea's freshwater budget. The latent heat fluxes associated with iceberg melt even seem to play a dominant role in salinity changes up to intermediate depths and the formation of deep water (Fig. 6 and 7). Furthermore, they lead to surface cooling in the Weddell Sea (Fig. 4b and h), which is also found by Stern et al. (2016). The altered spatial freshwater distribution alone leads to a warming of large areas of the

Southern Ocean's surface and subsurface waters (Fig. 4d) and thus buffers the cooling effect of iceberg melt. The iceberg-related heat fluxes are necessary to compensate for the surface warming and sustain the anomalous vertical heat transport.

A strengthening of AABW by up to 10% agrees well with findings by Jongma et al. (2009) and Martin and Adcroft (2010). ICB and $\text{ICB}_\text{HF}$ show similar strengthening of AABW in the Indo-Pacific basin and the most pronounced weakening of stratification in the Ross Shelf region, indicating the importance of the latent heat effect. Deep-water formation along continental shelves is a process commonly underestimated in CMIP6 models, whereas open-water deep convection is highly overestimated (Heuzé, 2021). Though a realistic representation of AABW formation along continental shelves is not feasible in our model setup due to spurious mixing along steep topography gradients. Our results aid to tackle the issue of open-ocean deep convection and emphasize the added value of a realistic representation of iceberg-related heat and freshwater fluxes in the Southern Ocean.

Our results indicate a cooling of deep-water masses in model runs with interactive icebergs. A pronounced cooling of the global deep ocean is recognized after around 200 years. The experiment that only considers heat fluxes from iceberg melting while using the default parameterized freshwater fluxes shows similar results as the fully coupled one including the iceberg-related meltwater. This result is not surprising as the same heat flux is applied to both simulations, leading to monotonous cooling. This cooling may aid to reduce deep ocean temperature biases as found for FESOM2 (Streffing et al., 2022; Sidorenko et al., 2019), and other climate models (e.g., Delworth et al., 2006, 2012; Jungclaus et al., 2013; Rackow et al., 2019; Sterl et al., 2012) (compare Fig. 2). Including model icebergs in the Northern Hemisphere may lead to even enhanced cooling as the total global negative latent heat flux applied to the ocean would be larger than in our setup presented here. The effect of Northern Hemispheric icebergs on AABW and AMOC strength, however, may be more complex as the heat and freshwater fluxes due to iceberg melting affect the density profile differently, and the location of iceberg melt plays an important role.

## 5  Conclusions

We have studied the effect of interactive icebergs on the surface and in particular deep-ocean water mass changes. We used a fully coupled ESM with higher resolution (up to $\sim 1/3°$) at continental shelf regions around Antarctica together with an interactive Lagrangian iceberg model (Rackow et al., 2017). The addition of the interactive iceberg model has a strong cooling impact at the surface (except in the Amundsen-Bellinghausen Seas) in our study, which can act to decrease typical warm sea surface temperature biases in the Southern Ocean of climate models. This cooling combined with freshwater forcing could considerably delay Southern Ocean greenhouse warming in climate projections (Schloesser et al., 2019). This effect can be expected to increase with increasing iceberg discharge and an associated increase in latent heat flux due to iceberg melting. Furthermore, it might also play a role in explaining the observed lack of a multi-decadal decrease in Antarctic sea ice (Rackow et al., 2022). The region of strongest warming after the inclusion of interactive icebergs (Amundsen-Bellinghausen Seas) is in remarkable agreement with the location of strongest observed warming around Antarctica. Therefore, our results could indicate a role for increased iceberg-related meltwater and heat fluxes in the observed warming. Interestingly, the addition of the iceberg model in our study leads to reduced deep-ocean temperatures in all ocean basins as well, where current climate

models have been shown to typically be too warm (Rackow et al., 2019). Originating in the upper layers of the Southern Ocean, the cooling effect propagates northward. Our results suggest that the latent heat flux from iceberg melting is the main driver for this large-scale cooling. Furthermore, our results show an increased salinity on the continental shelves around Antarctica due to northward freshwater export by northward-drifting icebergs. This results in enhanced deep-water formation along continental shelves, which is a process commonly underestimated by CMIP6 models that do not include a sophisticated treatment of iceberg-related meltwater and heat fluxes. Our results thus emphasize the importance of realistically representing iceberg-related heat and freshwater fluxes in the high southern latitudes not only for surface-related biases but also in order to reduce long-standing biases in deep-water formation.

Icebergs play a crucial role in maintaining a suitable heat and freshwater balance in coupled climate models. Originating from glaciers or ice shelves, icebergs transport vast amounts of fresh water into the surrounding ocean. When they melt, this freshwater is released, significantly affecting the distribution of salinity and temperature in the ocean. Additionally, icebergs serve as a sink for heat. As they melt, they withdraw heat from the surrounding ocean, resulting in local cooling. These two effects, the freshwater input and the cooling, alter water density and consequently affect the vertical mixing of water masses and the stability of the water column. These changes have far-reaching consequences for the heat distribution within the ocean, with implications for regional and global climate patterns.

In the current generation of coupled climate models, icebergs are commonly not yet incorporated to simulate these processes accurately, with few exceptions, e.g. Smith et al. (2021). Including icebergs in ESMs enables for a more accurate representation and feedbacks of ocean circulation patterns, the transport of heat, and the distribution of freshwater, contributing to improved understanding of past, present, and future climate change. The iceberg model aids in closing a gap between climate and ice sheet modeling. It allows for applications in a coupled climate-ice sheet model (like, for instance, used in Ackermann et al. (2020) or Niu et al. (2021)), enabling the simulation of highly dynamic periods of abrupt climate change like Heinrich Events. Icebergs will have a different impact on the global ocean circulation than coastal hosing in deglacial meltwater scenarios (Lohmann et al., 2020). It is therefore necessary to properly include interactive icebergs in climate models in order to examine relevant feedbacks. Applied in a bihemispheric setup, the model is an important tool in assessing teleconnections between the polar regions. Recent marine records from the Southern Ocean of iceberg-rafted debris provide a clear signal of ice-sheet dynamics and variability (Weber et al., 2014). Adequate iceberg simulation will open a new avenue for interpreting deglacial meltwater and iceberg decay. Furthermore, the proposed enhanced configuration of AWI-ESM2.1 with reduced biases at the surface and in the deep ocean is a good candidate for better climate projections, as it includes a novel model component that can impact the timing of Southern Ocean greenhouse warming and Antarctic sea ice decline and thus ultimately projections of ice sheet retreat and global sea level rise.

*Code availability.* FESOM2 is a free software and available from this site https://github.com/FESOM/fesom2. The version with interactive icebergs used in this study is available at this site https://github.com/ackerlar/fesom2/tree/icb_for_merge. ECHAM6, which is the atmosphere model of the MPI-ESM, is a property of the Max Planck Institute for Meteorology. Its model code is available at https://

code.mpimet.mpg.de/login?back_url=https%3A%2F%2Fcode.mpimet.mpg.de%2Fprojects%2Fmpi-esm-users%2Ffiles after registration at this site https://code.mpimet.mpg.de/projects/mpi-esm-license. The version used in this study is available at https://gitlab.awi.de/paleodyn/

Models/echam6/-/tree/6.3.05p2-awiesm-2.1_icb. The esm_tools version used in this study is available at https://github.com/esm-tools/esm_tools/tree/feat/levante_icebergs.

## Appendix A:  Size classes and scaling factors

**Table A1.** Scaling factors for iceberg experiments performed in this study.

| Area $A$ [km$^2$] | Scaling factor |
| --- | --- |
| $A <= 0.1$ | 100 |
| $0.1 < A <= 1$ | 50 |
| $1 < A <= 10$ | 10 |
| $10 < A <= 100$ | 1 |
| $A > 100$ | 1 |

The reference iceberg height $h_{ref}$ for deriving the calving area flux $A_{tot}$ from the calving volume flux $V_{tot}$ is set to 250 m:

$$A_{tot} = V_{tot}/h_{ref} \tag{A1}$$

400    The number of icebergs $N$ is derived from the total calving area by subtracting with a reference iceberg area $A_{ref}$, here the median of the power law distribution:

$$N = A_{tot}/A_{ref} \tag{A2}$$

with $A_{ref} = 2^{1/(k-1)}x_{min}$.

## Appendix B:  Estimation of heat budget

405    The global ocean cooling $\Delta T$ due to latent heat fluxes from iceberg melting $Q_l$ is estimated by:

$$\Delta T = \frac{Q_l}{c_{p,oce}m_{oce}} \approx 0.01 \ K \ century^{-1} \tag{B1}$$

with the heat capacity of seawater $c_{p,oce} = 3850$ kJ kg$^{-1}$ K$^{-1}$ and the global ocean mass $m_{oce} = 1,4 \cdot 10^{18}$ kg. The latent heat flux from iceberg melting is given by $Q_l = m_{disch}h_L$ with the total discharge flux $m_{disch}$ of $1,731$ Gt year$^{-1}$ and the enthalpy of fusion of ice $h_L$ with 334 kJ kg$^{-1}$.

*Author contributions.* LA ported the iceberg model code from FESOM1 to AWIESM2, with support from TR and KH. LA coded the iceberg seeding routine, performed the simulations, and analyzed and visualized the results. All authors contributed to the manuscript and the discussion of the results.

*Competing interests.* The authors declare that they have no conflict of interest.

*Acknowledgements.* Thanks go to the Max-Planck Institute in Hamburg (Germany) and colleagues from the Alfred-Wegener Institute (AWI) for making ECHAM6-JSBACH and FESOM available to us. The simulations presented in this study are performed using the esm-tools (Barbi et al., 2021). Model simulations were performed on the high-performance computer "Levante" of the German Climate Computing Center (Deutsches Klimarechenzentrum, DKRZ). For the postprocessing and plotting of FESOM data, the python packages $\mathrm{pyfesom2}$ (https: //pyfesom2.readthedocs.io/en/latest/) and $\mathrm{tripyview}$ (https://github.com/FESOM/tripyview) were used. We thank Robert Marsh and one anonymous reviewer for their constructive feedback and comments. The work was supported by the research topic "Ocean and Cryosphere under climate change" in the program "Changing Earth – Sustaining our future" of the Helmholtz Society. LA acknowledges funding from the Federal Ministry for Education and Research initiative PalMod (project PalModII 1.3, BMBF grant no. 01LP1917A). TR acknowledges support from the European Commission's Horizon 2020 collaborative project NextGEMS (grant number 101003470).

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

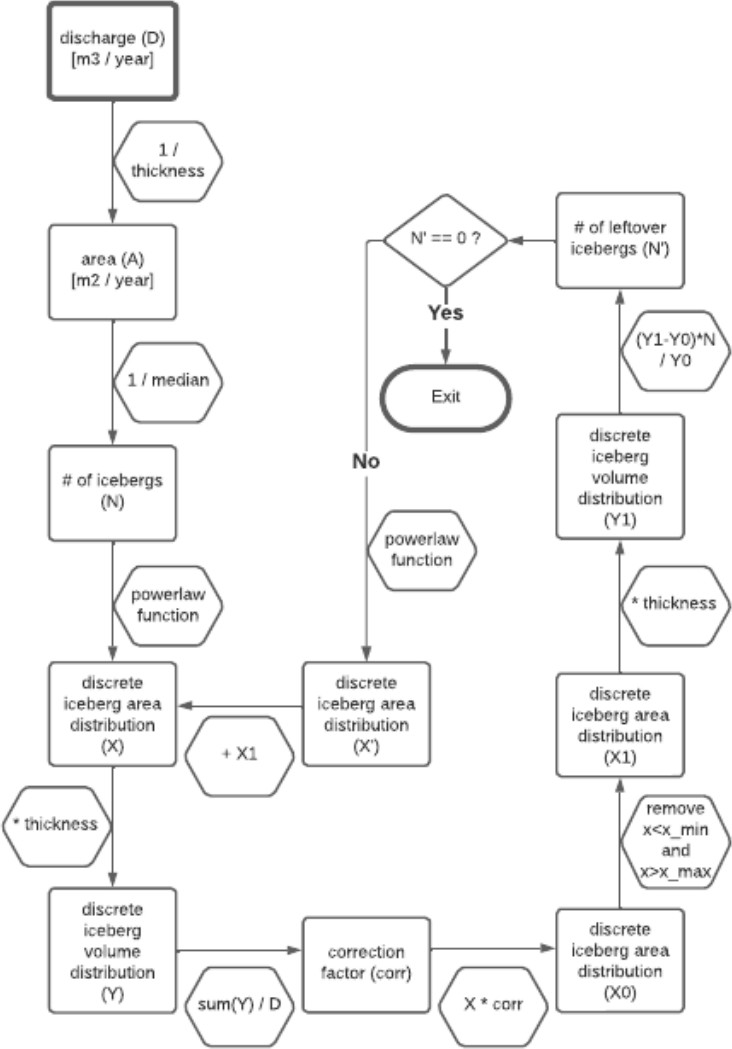

**Figure A1.** Scheme for the generation of discrete icebergs. The discharge field is integrated to receive a total discharge flux (D) which is divided by a reference iceberg height of 250 m to receive an iceberg area flux (A). A reference iceberg area size, here the median of the power-law distribution, is used to derive the number of icebergs to be generated (N). The median is given by $2^{1/(k-1)}x_{min}$ where k is $-3/2$ and $x_{min}$ is $0.01$ km$^2$. The number of icebergs N and the minimum area size $x_{min}$ are used with the $\mathrm{Python}$ power-law package to generate N discrete model icebergs with area size X. To compare the generated iceberg volume (Y) with the prescribed total discharge, X is multiplied by the reference ice thickness and summed up, to derive a scaling factor (corr). X is scaled with this factor to calculate a discrete area size distribution (X0) that is consistent with the prescribed total discharge. Those icebergs with areas smaller $x_{min}$ or larger $x_{max}$ are removed. For the total amount of removed iceberg volume, a new number of icebergs to be generated is calculated (N'). If this is zero, no further model icebergs are needed, i.e. the calculated model icebergs sum up to the given total discharge. If N' does not equal zero, an iterative process is started, in which new model icebergs from a power-law distribution a generated until N' is zero.

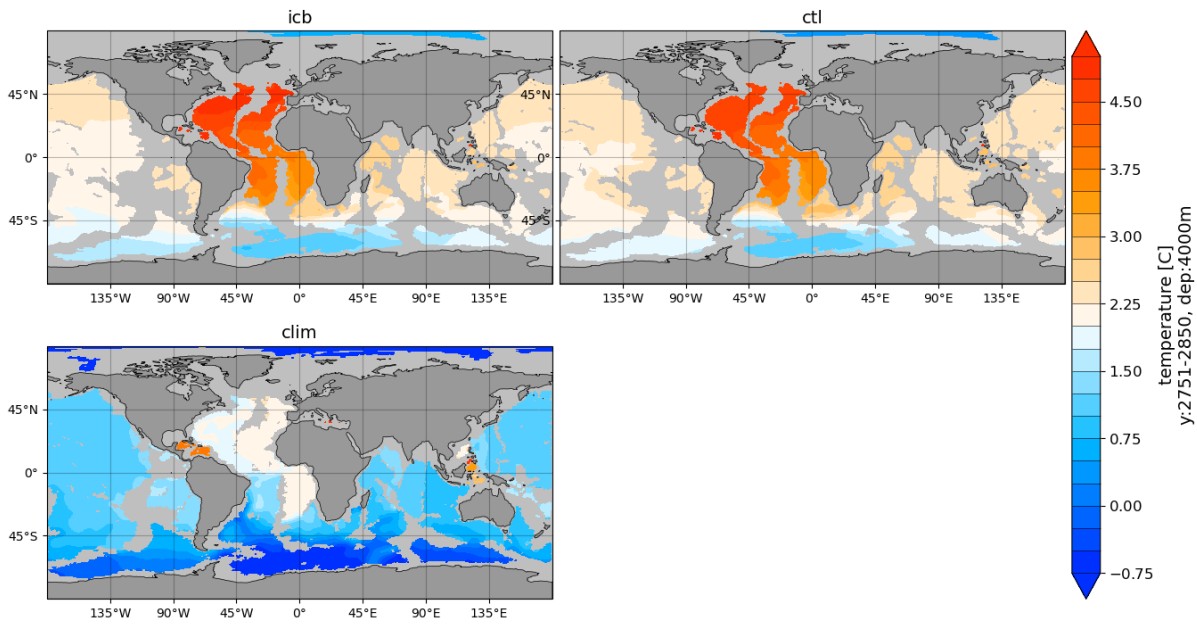

**Figure A2.** Temperature in 4,000 m depth compared to PHC3.0 for the last 100 model years of CTL and ICB.

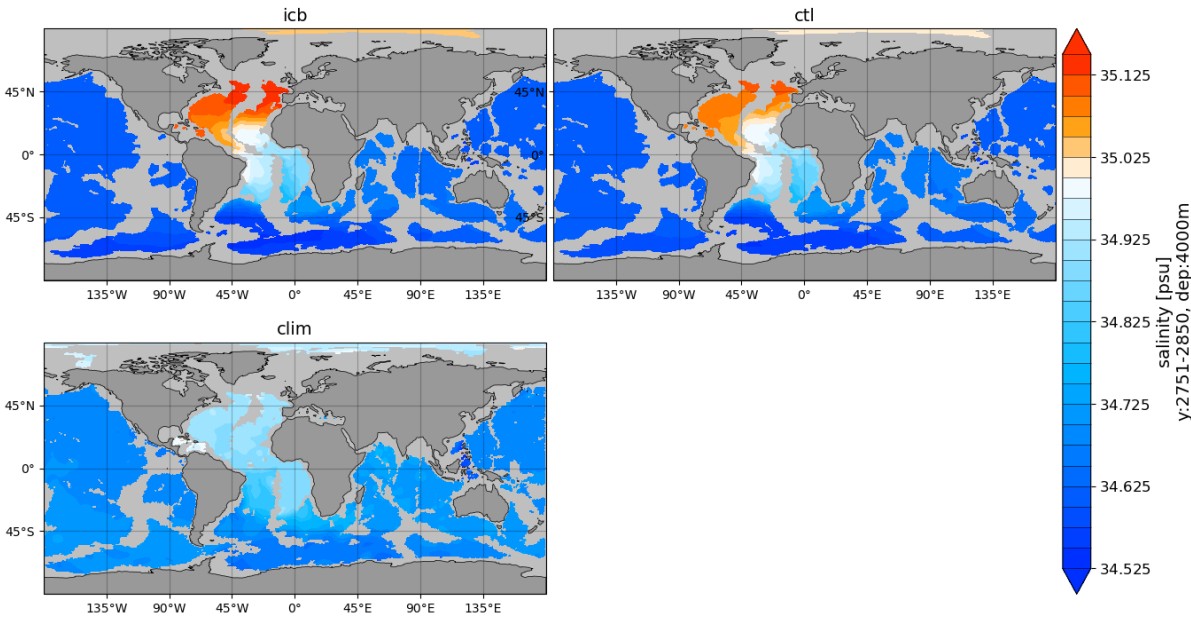

**Figure A3.** Salinity in 4,000 m depth compared to PHC3.0 for the last 100 model years of CTL and ICB.

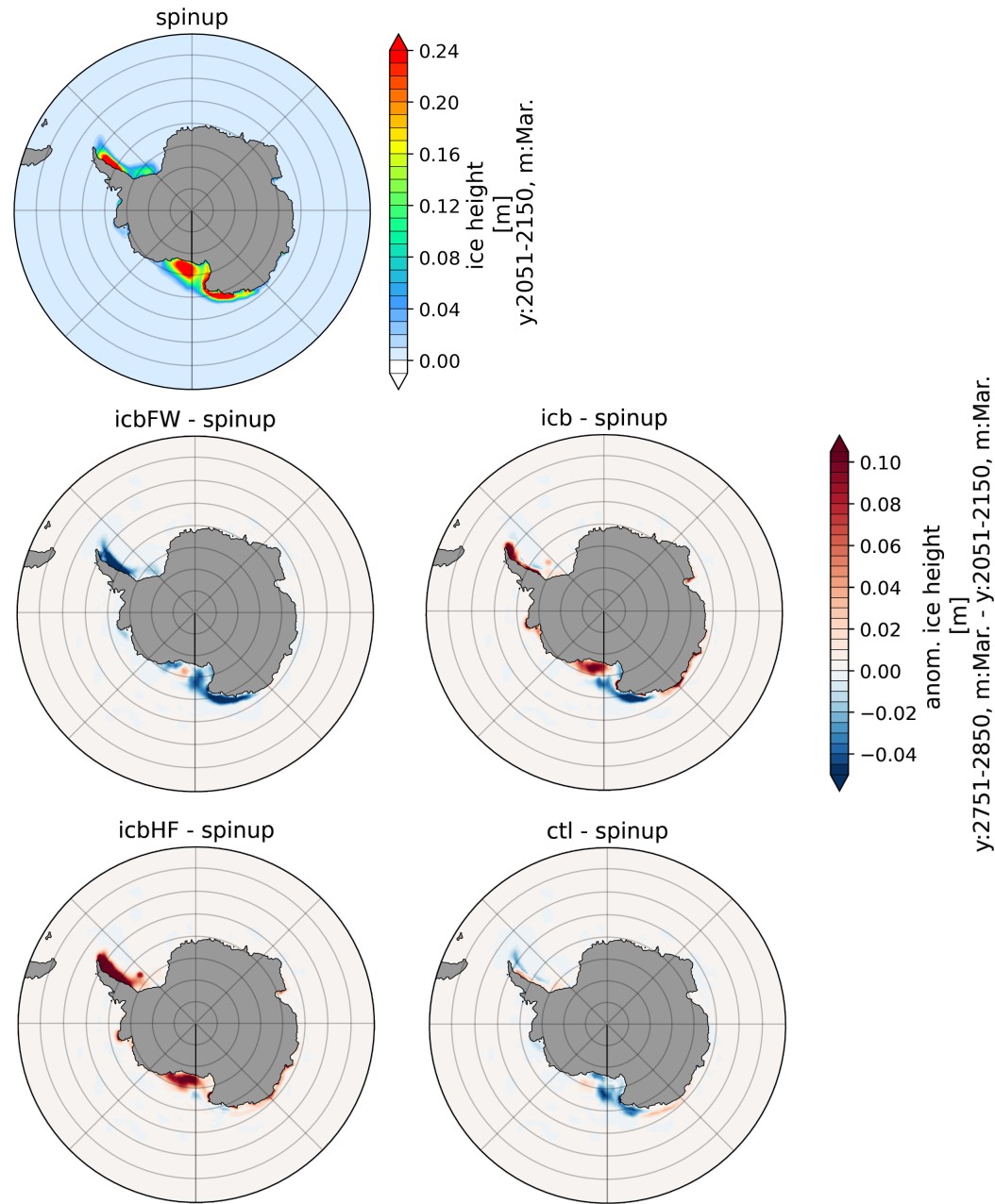

**Figure A4.** Anomaly of sea ice height for March of CTL, $ICB_{HF}$, $ICB_{FW}$, and ICB with respect to spinup for the last 100 model years.

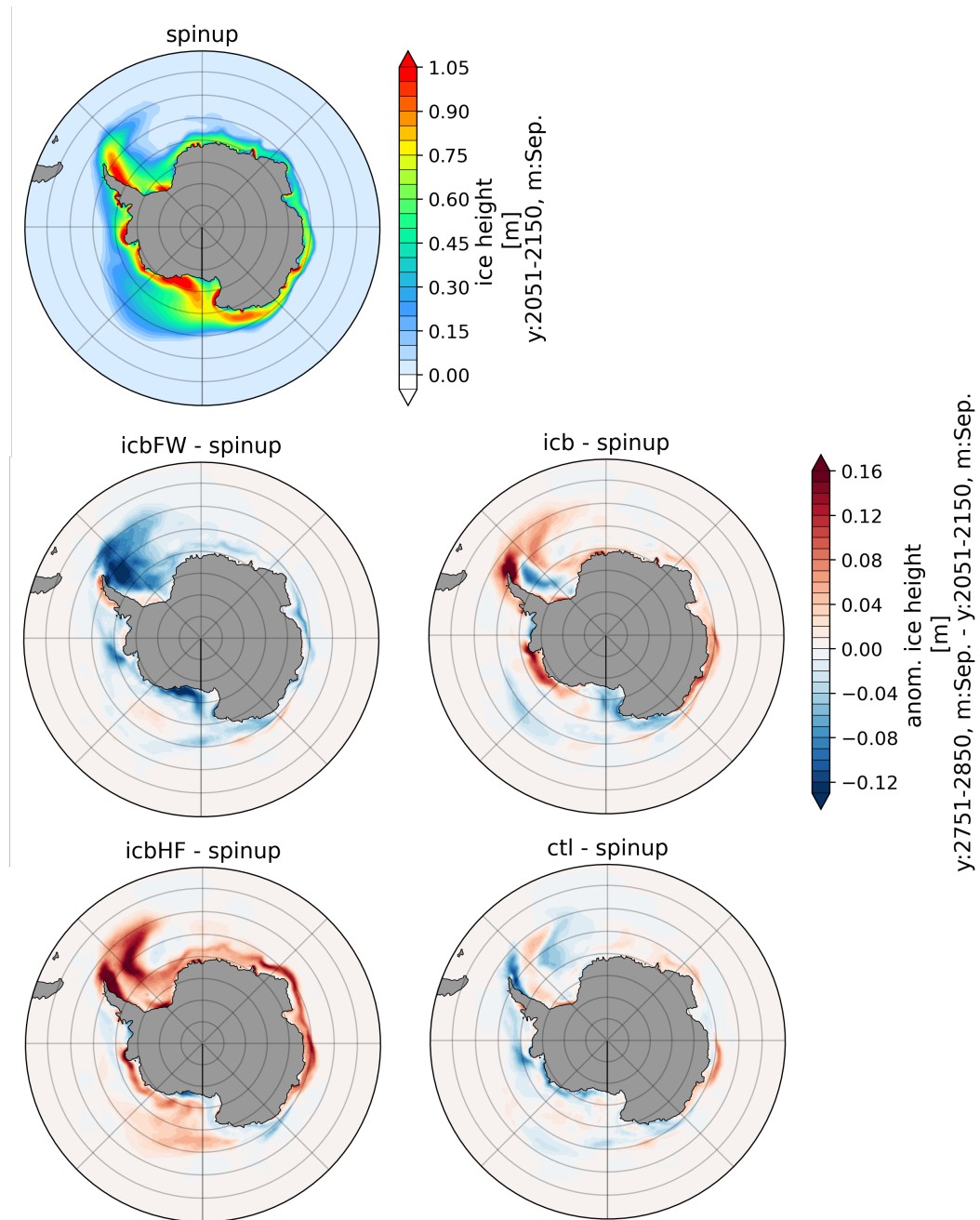

**Figure A5.** Anomaly of sea ice height for September of CTL, ICB$_{\text{HF}}$, ICB$_{\text{FW}}$, and ICB with respect to spinup for the last 100 model years.

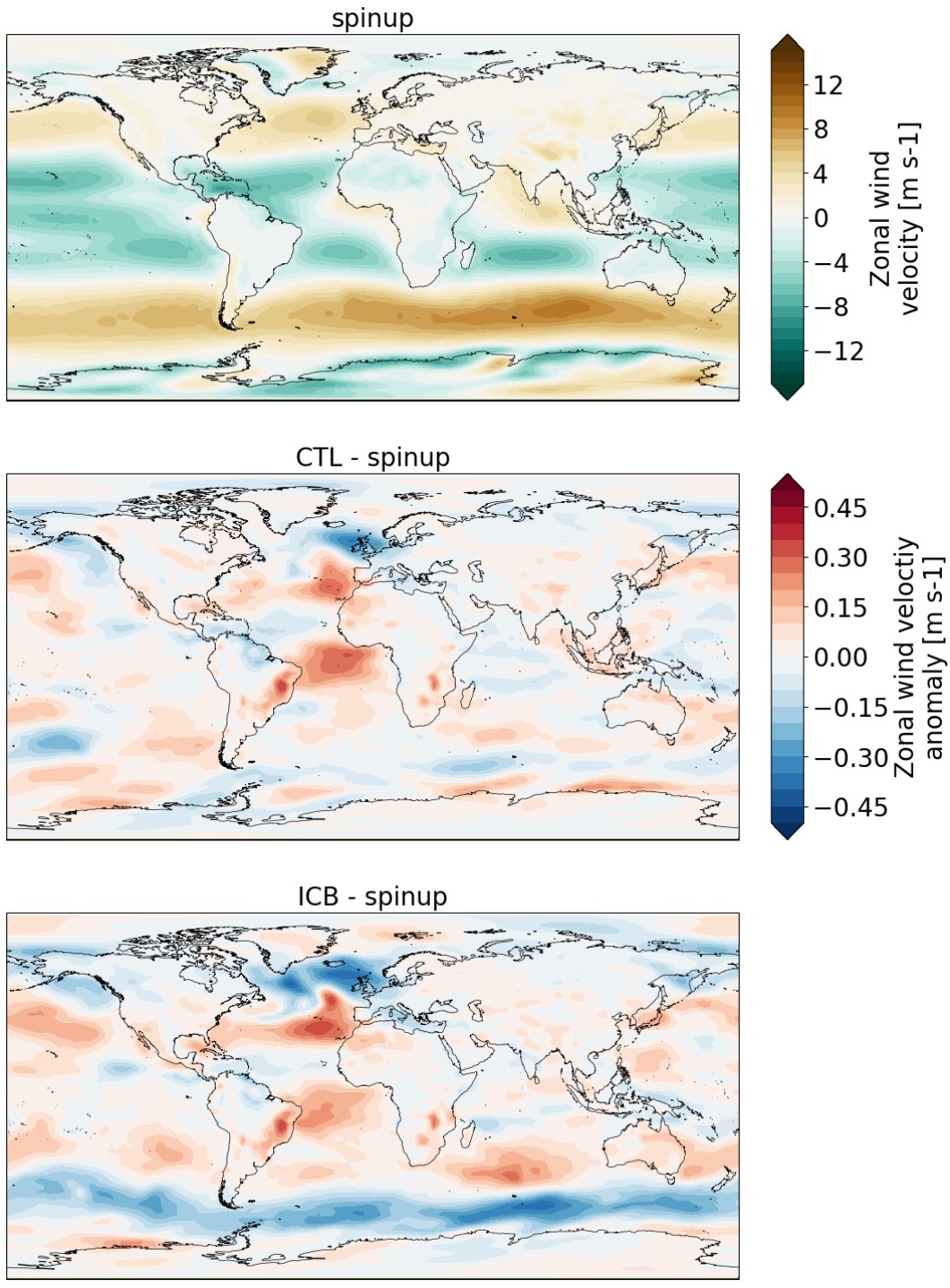

**Figure A6.** Zonal wind speed averaged over the last 100 model years for spinup, and zonal wind speed anomalies of CTL and ICB with respect to spinup.

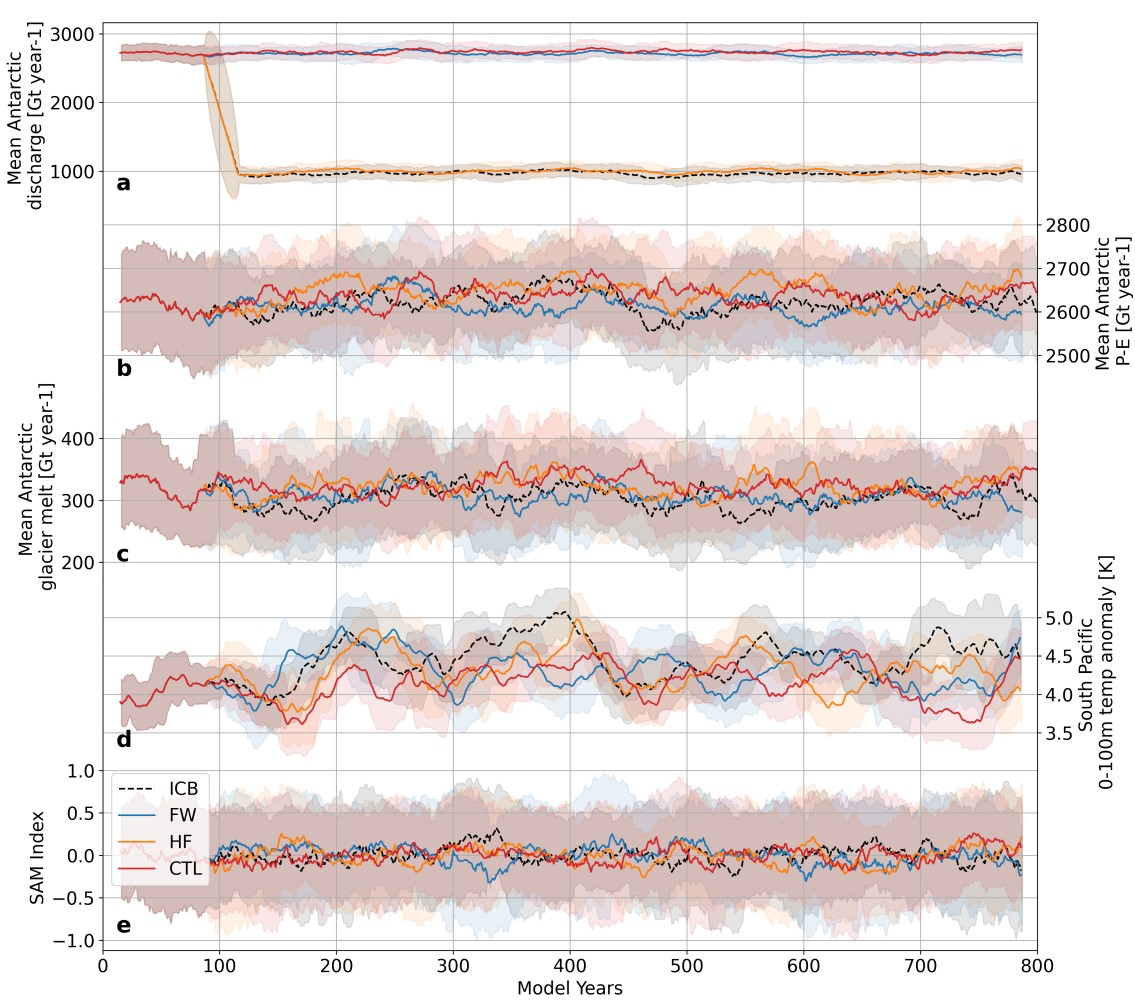

**Figure A7.** 50-year running mean timeseries of a) discharge from Antarctica into the ocean, b) precipitation (including snowfall) minus evaporation over Antarctica, c) glacier melt, d) temperature averaged over the upper 100 m between $110°$W-$130°$W and $55°$S-$65°$S, e) SAM Index calculated as the difference between the normalized monthly zonal sea level pressure at $40°$S and $65°$S of CTL, ICB, $\mathrm{ICB_{HF}}$ and $\mathrm{ICB_{FW}}$; shaded areas indicate one standard deviation.

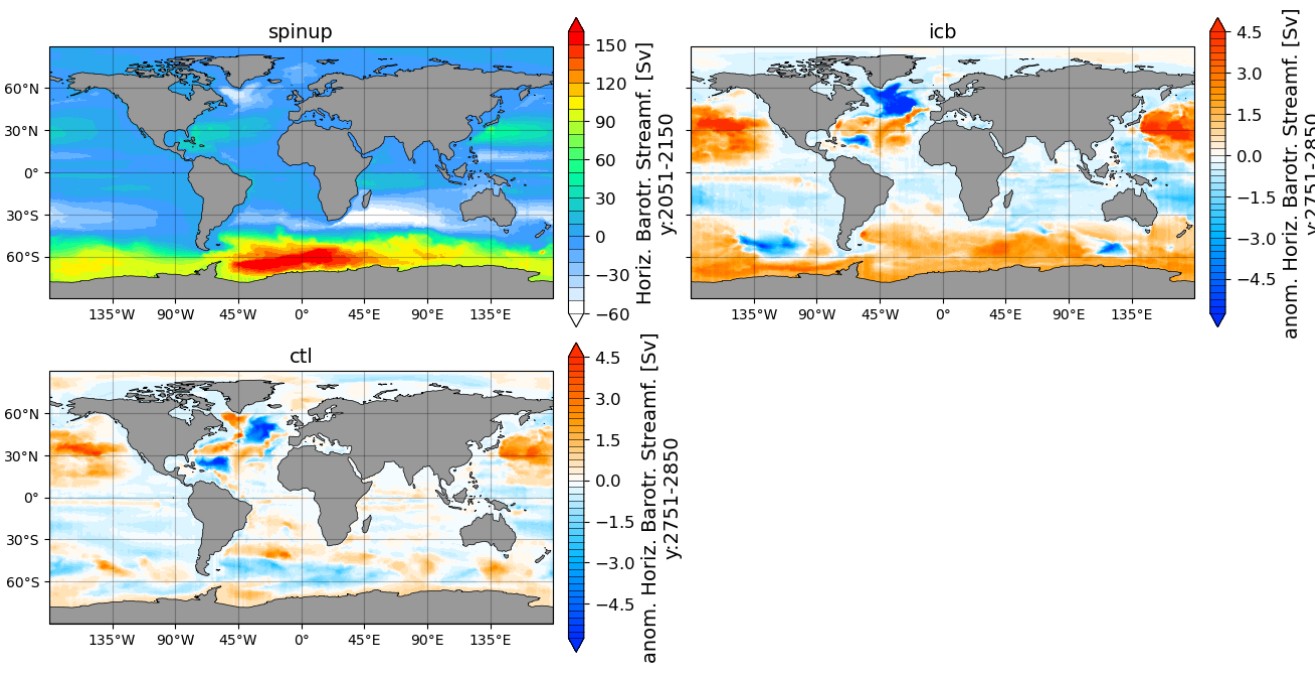

**Figure A8.** Barotropic stream function anomaly of CTL and ICB with respect to spinup for the last 100 model years.

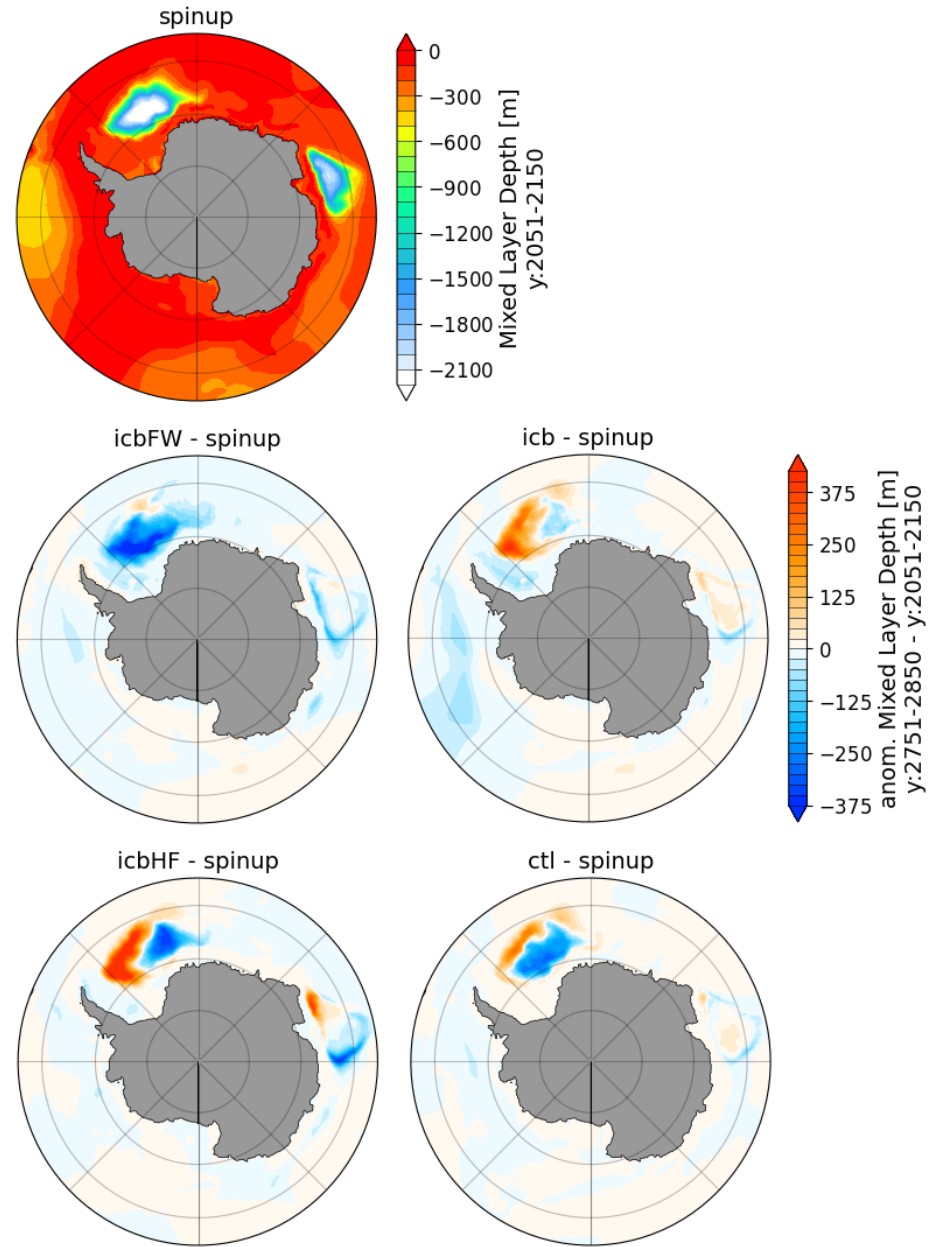

**Figure A9.** Mixed layer depth anomaly of CTL, ICB$_{HF}$, ICB$_{FW}$, and ICB with respect to spinup for the last 100 model years.

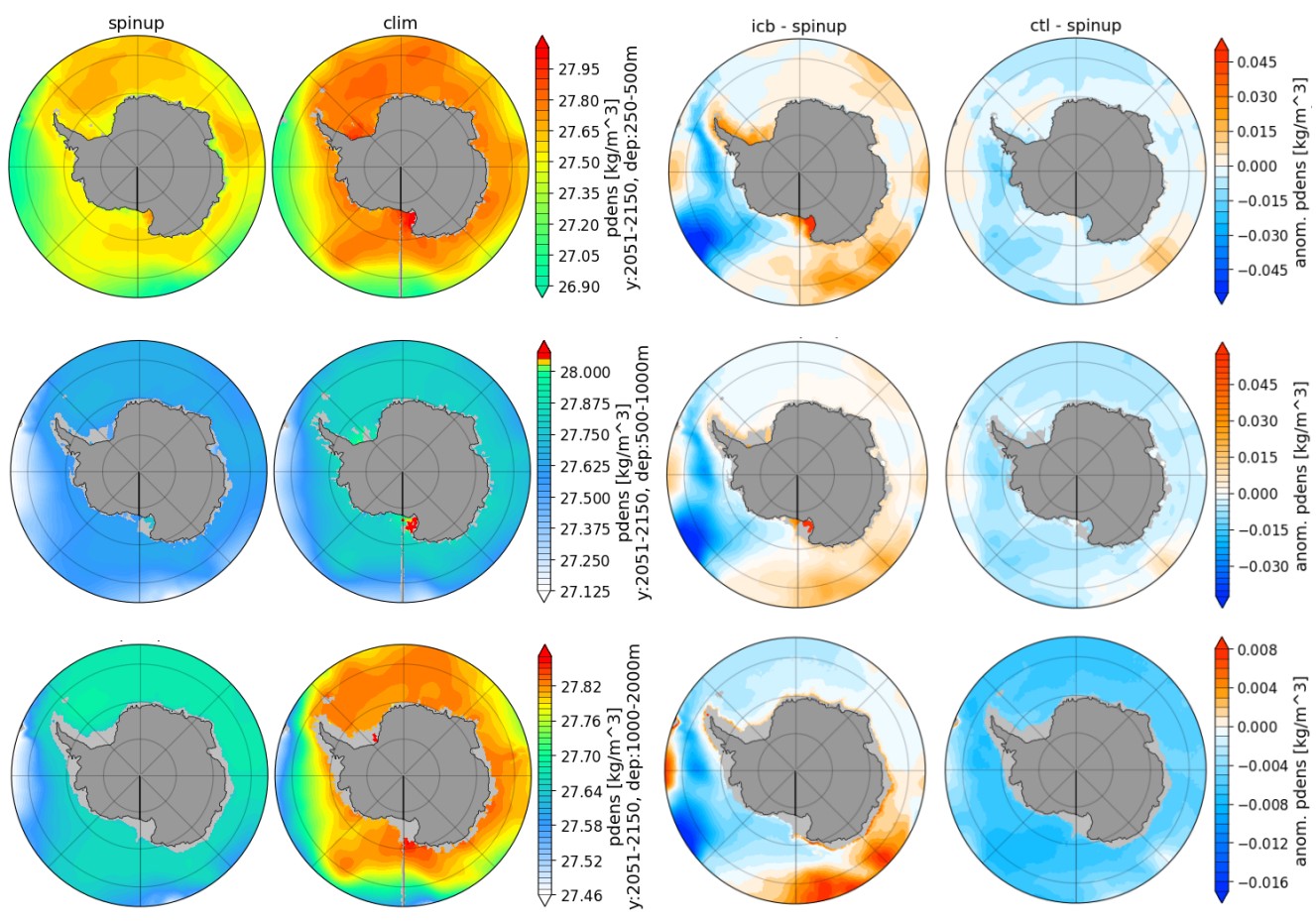

**Figure A10.** Climatology and anomaly of potential density of CTL and ICB at 250-500 m, 500-1,000 m, and 1,000-2,000 m with respect to spinup for the last 100 model years.