# Peer review of "A comprehensive Earth System Model (AWI-ESM2.1) with interactive icebergs: Effects on surface and deep ocean characteristics"

_EGUsphere, 2023_

## Referee Comment (RC1)

**General comments:**

I am very pleased to see a paper describing how an icebergs model can be implemented in an Earth System Model (ESM) and what are the effects on the oceanic mean state. This is clearly an important contribution that is relevant for the climate modelling community and for other groups that plan to do the same.

When interactive icebergs are introduced into an ESM, one of the critical aspects to address is the management of Antarctica runoff. It's worth noting that not all readers may be familiar with the surface runoff scheme used in the AWI ESM for Antarctica. Therefore, I believe it would be beneficial to begin by describing the existing AWI ESM surface runoff scheme for Antarctica and how the four components of Antarctica fresh water flux (surface mass balance, surface runoff, ice shelf basal melt, and icebergs) are managed it term of both fresh water and latent heat flux. Then add a more detailed explanation of the changes that occur once interactive icebergs are incorporated into AWI ESM. This information could prove relevant to other research groups that plan an integration of interactive icebergs into their ESMs.

There appears to be a gap in the description of the experimental runs. I found it challenging to understand how the reference 'surface' runoff is adjusted once interactive icebergs are introduced (or part of it like in $ICB_{HF}$ or $ICB_{FW}$), particularly concerning the heat fluxes. Throughout the manuscript, I had an ongoing question: whether the quantities of heat fluxes (specifically latent heat) and freshwater fluxes remain consistent across the various experiments, with the differences being attributed solely to the locations where the icebergs melt. This ambiguity in the experimental setup has made it difficult to fully grasp certain results and their implications. Clarifying this aspect would greatly enhance the understanding of the study.

Because of missing information about realism of the simulation, I encountered difficulties in assessing the applicability of your results to the real world or to other climate models. To address this concern, I suggest including a brief validation section focusing on the Southern Ocean. This validation could include key indicators such as gyre strength and extent, the Antarctic Circumpolar Current (ACC), sea ice extent, bottom temperature and salinity (S), and atmospheric circulation.

Furthermore, because you use an ESM, it would be beneficial to go a bit further in the analysis by investigating how the introduction of interactive icebergs influences these key indicators and the resulting feedback on atmospheric circulation, Antarctic surface mass balance, and surface runoff.

**Style and Figures:**

I found the paper to be well-written. The selection of figures is appropriate. The quality of the figures is high.

**Specific comments:**

Here are some specific comments.

**Section 1:**

- L57-67: Much of this paragraph appears to resemble the 'model description.' Please consider moving it, either in full or in part, to Section 2.

**Section 2:**
- L81: Add a reference to 'COREII' mesh. For many readers, COREII is more an atmospheric forcing set than a mesh. So, it could be worth adding some precisions here or a reference.
- L92-95: Could you also clarify whether there is a surface mass balance at the air/iceberg interface and specify the temperature, salinity, and velocity values used to calculate the different types of melt (surface or in-depth values)?
- L97-98: Do the icebergs remain stationary if they enter a saturated cell, or is only their velocity towards the saturated cell set to 0? What happens if calving takes place within a saturated cell? Are icebergs relocated away from the saturated cell?
- L98: Is there an iceberg grounding scheme? If yes, how this scheme works?
- L105-115: I believe this paragraph would be more appropriately placed at the end of line 135, where the total calving is mentioned.
- L109: What is the rationale behind fixing the compensation of freshwater constant in time as opposed to having it updated at the iceberg time step frequency depending of the instantaneous melt?
- L110: You balanced the discrepancy between the iceberg melt and the constant fresh water removed from the total runoff at every time step (?). Could you provide specific details on how this correction was implemented? Was the constant correction spread evenly worldwide or confined to the Antarctic runoff area?
- L110/L134: Your constant is the annual total calving flux (1,731 Gt/year). I am curious about the necessity for this correction. When examining your total water budget, it should be balanced at each time step if you consider the iceberg volume and the ice within the buffer used to replenish the category before icebergs are released within your model. Could you provide some insight into why the correction is still required in this context?
- L116: In the introduction, you mentioned Greenland but not later in the paper. Please add a sentence somewhere explaining that the study focuses solely on Antarctic icebergs and provide the reasoning behind this choice.
- L120-121: and Figure 1: The calving rate map appears unrealistic in certain areas. For instance, there is calving within the Ross Ice Shelf along the Trans-Antarctic Mountains, calving along the grounding lines of Getz and Amery, but no calving at Pine Island Glacier or Thwaites Glacier ice front. Additionally, there is calving almost at the Weddell Sea continental shelf break and hundreds of kilometers away of the coast line in Amundsen sea. These discrepancies raise questions about the accuracy of the calving rate data.
- L120: Could you provide more details about the highest calving rate mentioned in the paper? Does it refer to an entire ice shelf or to a particular point? If it is the second, the reported rate appears exceptionally high, almost at the level of an entire ice shelf, which requires further clarifications.
- L120-121: Is the calving rate assumed to be spatially constant within each sector by the model? If so, does this imply that there is effective calving of icebergs of one category in the model for all cells within the same sector simultaneously?

- L125: In the work by England et al. (2020), they use a similar iceberg size for the largest tabular icebergs and the same power law for the distribution. Therefore, it's challenging to understand how the iceberg melt pattern in your model could be realistic without considering the possibility of fragmentation.
- L127-128: I don't see how the prescription of iceberg thickness described in L126 prevent iceberg to be instantaneously grounded when created.
- L130: Could you give more details on the iterative process?
- L130: You described the iceberg size distribution by area bin. Could you mention what is the size of the icebergs released by the model for each category? If it is a very different method (size randomly generated, …), please give extra details because I found unclear how the effective iceberg size in the model is chosen.
- L130: reference is Fig. 1b not 1a.
- L132: Could you give precisions of where the calving occurs in the model by adding reference about which coastline you used. Is it the same as in Figure 1? If yes, how the discrepancy between observed and PISM coastlines (and thus calving location) affect the overall results?
- Table 1: Add units for 'run length' column and 'cpl. Frequency' column. Furthermore, it could be worth adding clarification about how the iceberg fwf and heat fluxes are managed for each simulation in the table with something like: interactive / absent / surface runoff for heat and fresh water iceberg fluxes.
- Figure 1b: I can only reproduce your number here by using a normalized power law distribution like the Pareto distribution. Furthermore, when I integrate the one in the appendix from $x_{min}$ to infinity, I have more than 1. I found this weird for a distribution.
- Appendix A: I am surprised to see case A>1000 km$^2$ because you mentioned in section 2 that the maximal iceberg size is 400 km$^2$.

**Section 3:**
- L146-152: When you begin to describe the differences between observed and modeled icebergs, it would be pertinent to mention that the modeled icebergs tend to be overly confined along the coast and do not exhibit the behavior of escaping the current along the Kerguelen Plateau.
- L174: Do you have insights into the reasons for the increase in salinity offshore in the Amundsen Sea? Is it a direct effect of icebergs, or does it result indirectly from changes in mean circulation or sea extent?
- Figure 3: revert colorbar (red > 0, blue < 0).
- Figure 3,4,5: To ensure the relevance of the processes in other climate model and 'real' world, it would be helpful to have the CTL (or ICB) fields displayed alongside the World Ocean Atlas 2018 fields, potentially in the initial comments of Section 3
- L199: As you pointed out, salinity plays a crucial role in controlling seawater density around Antarctica. Additionally, you mentioned in line 162 that icebergs do not melt on-site but further north. Given these considerations, I'm somewhat surprised by the relatively small differences in ICB$_{FW}$ compared to ICB$_{HF}$ on the Antarctic continental shelves, and on the modeled increased stratification in Figure 6. Intuitively, I would have expected the opposite effect. Could you please provide clarification on this discrepancy?

**Section 4:**

- L215-217: You briefly compared model icebergs to quickscat in section 3.1. Probably you should move this to section 3.1.
- L220-224: Fragmentation is missing in your model. You should mention this here.
- L229: Add a reference for 'size distributions vary at different location'.

**Section 5:**
- L287: Some ESM already contains an active Lagrangian iceberg model like UKESM.

---

## Author Response (AR1)

Reviewer 1:

General comments:

I am very pleased to see a paper describing how an icebergs model can be implemented in an Earth System Model (ESM) and what are the effects on the oceanic mean state. This is clearly an important contribution that is relevant for the climate modelling community and for other groups that plan to do the same.

When interactive icebergs are introduced into an ESM, one of the critical aspects to address is the management of Antarctica runoff. It's worth noting that not all readers may be familiar with the surface runoff scheme used in the AWI ESM for Antarctica. Therefore, I believe it would be beneficial to begin by describing the existing AWI ESM surface runoff scheme for Antarctica and how the four components of Antarctica fresh water flux (surface mass balance, surface runoff, ice shelf basal melt, and icebergs) are managed it term of both fresh water and latent heat flux. Then add a more detailed explanation of the changes that occur once interactive icebergs are incorporated into AWI ESM. This information could prove relevant to other research groups that plan an integration of interactive icebergs into their ESMs. There appears to be a gap in the description of the experimental runs. I found it challenging to understand how the reference 'surface' runoff is adjusted once interactive icebergs are introduced (or part of it like in ICBHF or ICBFW), particularly concerning the heat fluxes. Throughout the manuscript, I had an ongoing question: whether the quantities of heat fluxes (specifically latent heat) and freshwater fluxes remain consistent across the various experiments, with the differences being attributed solely to the locations where the icebergs melt. This ambiguity in the experimental setup has made it difficult to fully grasp certain results and their implications. Clarifying this aspect would greatly enhance the understanding of the study.

Because of missing information about realism of the simulation, I encountered difficulties in assessing the applicability of your results to the real world or to other climate models. To address this concern, I suggest including a brief validation section focusing on the Southern Ocean. This validation could include key indicators such as gyre strength and extent, the Antarctic Circumpolar Current (ACC), sea ice extent, bottom temperature and salinity (S), and atmospheric circulation. Furthermore, because you use an ESM, it would be beneficial to go a bit further in the analysis by investigating how the introduction of interactive icebergs influences these key indicators and the resulting feedback on atmospheric circulation, Antarctic surface mass balance, and surface runoff.

We would like to thank you for the careful reading and the timely and very constructive feedback.

Indeed, ice shelf basal melting and iceberg discharge are not considered explicitly in the default AWI-ESM version. All freshwater entering the ocean is considered surface runoff and treated by the hydrological scheme of the land-surface component. This implies that no latent heat fluxes due to iceberg melt are considered in the default model version. Hence, the two model versions, with and without interactive icebergs, are not identical with respect to the heat fluxes. While the individual patterns of freshwater release are different in both model approaches, the sum of the freshwater fluxes is identical, as the surface runoff from Antarctica (that implicitly includes basal shelf melt fluxes and iceberg calving) is reduced by the amount of freshwater input due to calving in the model version with interactive icebergs. The model icebergs melt, eventually, and release this fresh water into the ocean.

A paragraph about the surface runoff and snow melt treatment in ECHAM/JSBACH is added to the model description section (L746-83). Snow melt is not considered on glacial areas, and all excess snow is just added to the river routing scheme as liquid freshwater and eventually released to the ocean at the estuaries. Hence, the iceberg discharge and basal melting of Antarctica are only implicitly accounted for.

With regard to the realism of the simulation, we include zonal mean (Fig. 2) and 4,000 m temperature and salinity deviations of CTL from PHC3.0 (Fig. A2 and A3). Strong warm biases are present at the deep ocean and fresh biases on thee continental shelves (Fig. 2). Furthermore, we see too warm and too fresh water masses in the deep ocean (Fig. A2 and A3), resulting in too light waters at different depth levels (Fig. A10). These biases indicate that AABW is not formed by overflow along steep topographies but rather by open-ocean convection (Fig. A9).
Sea ice cover is underestimated in the Weddell Sea and the Atlantic sector during summer and winter seasons, and in the Amundsen and Bellinghausen Seas during the summer (Fig. A4 and A5). Timeseries for the ACC and Weddell Gyre strength are included (Fig. 8). The Drake Passage throughflow is around 87-90 Sv in CTL, which is significantly smaller than suggested by observations (Donohue et al., 2016; ,Whitworth et al., 1985). In our simulations, the Drake Passage throughflow increases to around 92 Sv in ICB. The Weddell Gyre strength, however, shows similar variability (multi-decadal to centennial) and amplitude (80-89 Sv) for all simulations. All changes are within one standard deviation.

We took a closer look at atmospheric feedback and Antarctica's mass balance. The southern westerlies are weakened in ICB compared to CTL, however, the effect is less than 10 % (Fig. A6). Regarding the surface mass balance of Antarctica, only the discharge shows significant changes (Fig. A7a). As mentioned in the methods section, the river runoff is artificially reduced by the amount of calving flux. The net precipitation (P-E) and glacial melting do not show siginificant changes (Fig. A7b and c).

Specific comments:

- L57-67: Much of this paragraph appears to resemble the 'model description.' Please consider moving it, either in full or in part, to Section 2.
- The paragraph has been moved to Section 2 (L90-95).

- L81: Add a reference to 'COREII' mesh. For many readers, COREII is more an atmospheric forcing set than a mesh. So, it could be worth adding some precisions here or a reference.
- The term „COREII" for the ocean mesh has been dropped, and references are added at L89.

- L92-95: Could you also clarify whether there is a surface mass balance at the air/iceberg interface and specify the temperature, salinity, and velocity values used to calculate the different types of melt (surface or in-depth values)?
- Except wind drag, no exchange processes with the atmosphere are implemented in the model. Compared to melting due to oceanic processes, the effect of surface melting due to radiation is minor and commonly neglected (Bigg et al., 1997). Temperature, salinity, and velocity values are either taken at the iceberg base (for basal melting), or averaged over the iceberg depth. An additional paragraph is included in the revised manuscript (L106-118).

- L97-98: Do the icebergs remain stationary if they enter a saturated cell, or is only their velocity towards the saturated cell set to 0? What happens if calving takes place within a saturated cell? Are icebergs relocated away from the saturated cell?
- If the new iceberg leads to a larger total iceberg area than the actual element area, it does not move but stays in its previous grid element and is set back to its previous position. Hence, it can still move within the grid cell element or to a neighboring grid cell that is not saturated yet. Furthermore, model icebergs are not discharged into saturated ocean grid cells but are distributed over the coastal and neighboring grid cells within the respective basin (Fig. 1). An additional paragraph is included in the manuscript (L119-124).

- L98: Is there an iceberg grounding scheme? If yes, how this scheme works?
- Whenever the model iceberg's depth reaches deeper than the local bathymetry, it is assumed to be grounded, and its velocity is set to zero. Basal melting can still occur and eventually set the model iceberg free again once its depth is sufficiently reduced. An additional paragraph is included in the manuscript (L124-126).

- L105-115: I believe this paragraph would be more appropriately placed at the end of line 135, where the total calving is mentioned.
- Thank you for this comment! The paragraph you refer to mentions new development with respect to the iceberg model itself and the coupling with FESOM. The calving rather constitues a coupling between the ice sheet model output and FESOM and is independent of the overall new implementation within the iceberg module. Hence, we assume this paragraph is better fitting in the „Iceberg module" section than in the „Iceberg seeding" section.

- L109: What is the rationale behind fixing the compensation of freshwater constant in time as opposed to having it updated at the iceberg time step frequency depending of the instantaneous melt?
- The compensation accounts for the actual calving, i.e. the reduction of surface runoff off Antarctica. This calving flux is constant in our simulation setup. The reduction in surface runoff is applied at every atmosphere-ocean coupling time step. However, the application in a coupled climate-ice sheet model would allow for a time varying calving flux and the corresponding runoff compensation would vary accordingly. The instantaneous melt is accounted for by the FESOM internal salinity balance. Hence, the total salinity is held constant. An additional paragraph is included in the manuscript (L137-145).

- L110: You balanced the discrepancy between the iceberg melt and the constant fresh water removed from the total runoff at every time step (?). Could you provide specific details on how this correction was implemented? Was the constant correction spread evenly worldwide or confined to the Antarctic runoff area?
- The discrepancy arises from the fact that not all discharged icebergs melt instantaniously, hence the discharge flux does not equal the actual iceberg melting flux. To account for this, the total salinty within the ocean domain is held constant. This also allows for better comparison with default model simulations without interactive model icebergs. The correction is spread evenly worldwide and is not confined to the Antarctic runoff area. Informtion were added at L137-145 (see above).

- L110/L134: Your constant is the annual total calving flux (1,731 Gt/year). I am curious about the necessity for this correction. When examining your total water budget, it should be balanced at each time step if you consider the iceberg volume and the ice within the buffer

used to replenish the category before icebergs are released within your model. Could you provide some insight into why the correction is still required in this context?

- The salinity is balanced at every time step. But there are two different freshwater fluxes to consider:

    1) the calving flux that as actually treated as a reduction in the surface runoff provided from the hydrological river model, and is constant in time. The effect of this surface runoff reduction is a local salinity increase along the Antarctic coast.

    2) the actual iceberg melting flux which varies in time and space, is a FESOM internal freshwater flux (the iceberg model is just a submodel of FESOM) and is compensated globally. The effect of this flux is a local salinity reduction. Hence, both fluxes together just lead to a redistribution of fresh water from the coast to the open ocean. Information were added at L137-145 (see above).

- L116: In the introduction, you mentioned Greenland but not later in the paper. Please add a sentence somewhere explaining that the study focuses solely on Antarctic icebergs and provide the reasoning behind this choice.
- While the model allows for interactive icebergs in both hemispheres, our simulations only include iceberg in the Southern Ocean. We acknowledge the potential implications of iceberg-related freshwater  and heat fluxes for deep-water formation in the North Atlantic and, hence, on AMOC. However, our main focus is an enhanced understanding of processes involved in climate iceberg interactions rather than simulating realistic climatologies. Some explanation is added at L62-65 + L330-334.

- L120-121: and Figure 1: The calving rate map appears unrealistic in certain areas. For instance, there is calving within the Ross Ice Shelf along the Trans-Antarctic Mountains, calving along the grounding lines of Getz and Amery, but no calving at Pine Islan Glacier or Thwaites Glacier ice front. Additionally, there is calving almost at the Weddell Sea continental shelf break and hundreds of kilometers away of the coast line in Amundsen sea. These discrepancies raise questions about the accuracy of the calving rate data.
- Yes, model data show some discrepancies. However, total calving fluxes are integrated (summed up) over each basin, and eventually, model icebergs are seeded along the coast so that local discrepancies average partly out. Furthermore, the study aims more for process understanding rather than for realistic simulations.

- L120: Could you provide more details about the highest calving rate mentioned in the paper? Does it refer to an entire ice shelf or to a particular point? If it is the second, the reported rate appears exceptionally high, almost at the level of an entire ice shelf, which requires further clarifications.
- The highest calving rates per grid cell are about 10 Gt/year on a 16x16 km grid or ~45 m/year. It has been corrected in the manuscript at L155f.

- L120-121: Is the calving rate assumed to be spatially constant within each sector by the model? If so, does this imply that there is effective calving of icebergs of one category in the model for all cells within the same sector simultaneously?
- Calving fluxes are integrated over each basin and no differentiation regarding iceberg size classes or size-distributions is made within a basin. However, iceberg sizes are drawn randomly from a prescribed size-distribution and distributed randomly along the coastline of each basin. So no, different cells within one basin (or sector) may see different size-distributions. Additional information is added in the manuscript (L172-175 + Fig. A1).

- L125: In the work by England et al. (2020), they use a similar iceberg size for the largest tabular icebergs and the same power law for the distribution. Therefore, it's challenging to

understand how the iceberg melt pattern in your model could be realistic without considering the possibility of fragmentation

• Giant icebergs are confined to coastal regions in our simulations. Once they cross the ACC, the icebergs are sufficiently small not to travel unrealistic long distances anymore. When looking at individual large icebergs, they tend to stay at a single location for several years. A feature facilitated by the parametrization to avoid saturation of ocean grid cells with model icebergs. This suggests that a break-up parametrization like in England et al. (2020) would not affect our results significantly, as most giant icebergs deteriorate before reaching the open Ocean, where the „footloose " mechanism would come into play. Furthermore, our maximum iceberg area is 400 km2, while icebergs > 400 km2 make up around 30% of freshwater flux in England et al. (2020). Some information werde added at L294-297.

• L127-128: I don't see how the prescription of iceberg thickness described in L126 prevent iceberg to be instantaneously grounded when created.
• You are correct, it does not prevent instantenous grounding but only reduces the risk for it. It has been clarified in the manuscript (L165).

• L130: Could you give more details on the iterative process?
• The discharge field is integrated to receive a total discharge flux (D) which is divided by a reference iceberg height of 250 m to receive an iceberg area flux (A). A reference iceberg area size, here the median of the powerlaw distribution, is used to derive the number of icebergs to be generated (N). The number of icebergs N and the minimum area size xmin are used with the python powerlaw package to generate N discrete model icebergs with area size X. To compare the generated iceberg volume (Y) with the prescribed total discharge, X is multiplied by the reference ice thickness and summed up, to derive a scaling factor (corr). X is scaled with this factor to calculate a discrete area size distribution (X0) that is consistent with the prescribed total discharge. Those icebergs with areas smaller xmin or larger xmax are removed. For the total amount of removed iceberg volume, a new number of icebergs to be generated is calculated (N'). If this is zero, no further model icebergs are needed, i.e. the calculated model icebergs sum up to the given total discharge. If N' does not equal zero, an iterative process is started, in which new model icebergs from a powerlaw distribution a generated until N' is zero. A new figure was added to illustrate the coupling scheme (Fig. A1 + caption). And a reference to Fig. A1 was added at L167.

• L130: You described the iceberg size distribution by area bin. Could you metion what is the size of the icebergs released by the model for each category? If it is a very different method (size randomly generated, …), please give extra details because I found unclear how the effective iceberg size in the model is chosen.
• Icebergs are drawn randomly from a powerlaw distribution (using the pyhon „powerlaw" package). Please see the response above and Fig. A1 for more details.

• L130: reference is Fig. 1b not 1a.
• Thank you, for pointing this out. It has been corrected in the manuscript (L168).

• L132: Could you give precisions of where the calving occurs in the model by addin reference about which coastline you used. Is it the same as in Figure 1? If yes, how the discrepancy between observed and PISM coastlines (and thus calving location) affect the overall results?
• For the seeding, a list of ocean grid cells is generated. For each ice sheet model grid cell in which calving occurs, the nearest ocean grid cell and the neighbouring cells are added to this list. For each basin, model icebergs are then distributed over the ocean grid cells, contained

in the list. No sensitivity runs with respect to calving location have been made. Information about this were added at L169-172.

- Table 1: Add units for 'run length' column and 'cpl. Frequency' column. Furthermore, it could be worth adding clarification about how the iceberg fwf and heat fluxes are managed for each simulation in the table with something like: interactive / absent / surface runoff for heat and fresh water iceberg fluxes.
- Thank you for this suggestion! The table has been adjusted accordingly (Table 2 (in the new manuscript) + caption).

- Figure 1b: I can only reproduce your number here by using a normalized power law distribution like the Pareto distribution. Furthermore, when I integrate the one in the appendix from xmin to infinity, I have more than 1. I found this weird for a distribution.
- You are correct, the power law distribution mentioned in Tournadre et al. (2016) should probably be

$$(\alpha-1) \cdot x_{min}^{1-\alpha}$$

instead of

$$\frac{\alpha-1}{x_{min}} \cdot x_{min}^{1-\alpha}$$

Hence, the distribution is scaled with $1/x_{min}$, i.e. a factor of 100 with $x_{min}=0.01$. However, this does not affect the iceberg generation in our setup, that follows the scheme shown in Fig. A1.

- Appendix A: I am surprised to see case A>1000 km2 because you mentioned in section 2 that the maximal iceberg size is 400 km2.
- Thank you, you are right. the table has been fixed.

- L146-152: When you begin to describe the differences between observed and modeled icebergs, it would be pertinent to mention that the modeled icebergs tend to be overly confined along the coast and do not exhibit the behavior of escaping the current along the Kerguelen Plateau.
- A sentence has been added in the manuscript (L193f).

- L174: Do you have insights into the reasons for the increase in salinity offshore in the Amundsen Sea? Is it a direct effect of icebergs, or does it result indirectly from changes in mean circulation or sea extent?
- The change in the buoyancy frequency affects the magnitude of vertical mixing (Fig. 7) that is reduced (enhanced) in the Weddell Sea for ICBHF and ICB (CTL and ICBFW). The increased vertical mixing in the Amundsen and Bellinghausen Seas in ICB leads to an upward heat transport that results in surface warming (Fig. 4a). Information were added at L254-257.

- Figure 3: revert colorbar (red > 0, blue < 0).
- The figure has been changed accordingly (Fig. 4 (in the new manuscript)).

- Figure 3,4,5: To ensure the relevance of the processes in other climate model and 'real' world, it would be helpful to have the CTL (or ICB) fields displayed alongside the World Ocean Atlas 2018 fields, potentially in the initial comments of Section
- New Figures are included (Fig. 2 (in the new manuscript), A2 and A3) that show zonal mean (Fig. 2 (in the new manuscript)) and 4,000 m depth anomalies of temperature and salinity for CTL with respect to the PHC3.0 (Steele et al., 2001). A brief description is added to the manuscript (L185-189).

- L199: As you pointed out, salinity plays a crucial role in controlling seawater density around Antarctica. Additionally, you mentioned in line 162 that icebergs do not melt on-site but further north. Given these considerations, I'm somewhat surprised by the relatively small differences in ICBFW compared to ICBHF on the Antarctic continental shelves, and on the modeled increased stratification in Figure 6. Intuitively, I would have expected the opposite effect. Could you please provide clarification on this discrepancy?
- In case of ICBFW, there are two competing processes: 1) the reduced surface runoff which would (with respect to CTL or the spinup) result in positive salinity anomalies; 2) less sea-ice formation and less brine rejection. In the Weddell Sea we have large calving rates, hence a strong reduction in surface runoff (Fig. 1a) which could explain the strong positive salinity anomaly (Fig. 4d (in the new manuscript)). In the Ross Sea, however, there is still a high calving rate but not as large as in the Weddell Sea. Furthermore, there is a strong freshwater influx from iceberg melting (Fig. 3b (in the new manuscript)) that might counteract the reduced surface runoff. In ICBHF, we see the effects of latent heat fluxes due to iceberg melting alone. These negative heat fluxes foster enhanced sea-ice formation (very pronounced in the Ross Shelf region during summer; and along the sea-ice edge during winter, Fig. A7 & A8). Due to this enhanced sea-ice formation, the model experiences more brine rejection. In ICB all these processes add up.

- L215-217: You briefly compared model icebergs to quickscat in section 3.1. Probably you should move this to section 3.1.
- The paragraph is left in the Discussion section as reasons for the model icebergs behaviour are discussed but deviations from model icebergs to observations are now also mentioned in the Results section (see above L193f).

- L220-224: Fragmentation is missing in your model. You should mention this here.
- The iceberg model, on the other hand, does not include a breakup parametrization for large icebergs. Hence, the occurence and longevity of large icebergs, might be overestimated when compared to observation. Information were added at L291f.

- L229: Add a reference for 'size distributions vary at different location'. Section 5.
- A reference to Qi et al. (2021) has been added at L300.

- L287: Some ESM already contains an active Lagrangian iceberg model like UKESM.
- The statement is weakened and the UKESM is mentioned now (L363).

Reviewer 2:

General Comments

The authors present a comprehensive evaluation of interactive icebergs in a set of experiments with the AWI-ESM2.1 Earth System Model, demonstrating an advance in modelling capability that is within the scope of GMD. Multi-centennial experiments are necessary to establish the impacts of icebergs on deep and bottom waters that circulate on these long timescales. To my knowledge, these are the longest ESM experiments undertaken with interactive icebergs, while maintaining suitably high resolution around Antarctica. The separate impacts of iceberg-altered freshwater and heat fluxes are central to understanding the overall impacts of icebergs on the ocean, the latter impact (cooling) being regrettably a late admission to iceberg-ocean coupling in NEMO-ICB. Prospects for coupling with an interactive ice sheet are further promising. Overall, this study provides an advance on previous studies that reported the first implementations and subsequent developments of interactive iceberg modelling, several published in GMD. Methods are clearly outlined, with scope for minor clarification (see specific comments below), and results are sufficient to support conclusions. Following Code Availability, all results should be reproducible with sufficient computing resource. The title clearly reflects the contents of the paper, identifying the ESM by name.

A minor issue is the focus on Antarctica, with Greenland briefly mentioned in the Introduction. For a more complete representation, interactive icebergs can presumably also be seeded around Greenland (and other smaller northern ice caps). I appreciate that Antarctica is the dominant source of icebergs at global scale, but dense water formation in the subpolar North Atlantic may be particularly sensitive to the same processes (perturbing T and S) that are identified around Antarctica. It would be useful for the authors to comment further in Introduction and Discussion on the global impact of icebergs, in particular the net results of perturbations to deep and bottom water formation in both hemispheres. The Abstract provides a clear and concise summary of findings, although more quantitative detail could be provided (see specific comments). Overall presentation is well-structured throughout, and the manuscript is well written, with appropriate level of Supplementary Material.

In summary, the authors provide a compelling case for a relatively straightforward model improvement, of consequence for climate simulation and projection, with the intriguing possibility that neglected appreciation of iceberg influences could explain recent observed variability around. Antarctica (lines 266-269). As a development and technical paper, the manuscript should be suitable for publication in GMD, subject to minor revisions as suggested above and specified below.

We would like to thank you for the interest in our study and the timely and very constructive comments. While the main focus of this study is an enhanced process understanding of the long-term effects of interactive icebergs, we acknowledge the importance of Northern Hemisphere icebergs on deep-water formation in the North Atlantic and, ultimately, on the AMOC and the climate state.

Specific Comments

- Abstract: Provide additional quantitative support for statements, e.g., how much cooling, extent of (%) weakening in stratification.
- Quantitative statements about the cooling the reduction in vertical stratification are added to the manuscript (L9 & L12-13).

- Introduction, from line 25: Given the focus here on a new model capability in AWI-ESM2.1, and earlier comment in the Abstract that 'only few advanced coupled Earth System models that employ interactive icebergs', it would be appropriate to more comprehensively summarize these models, perhaps tabulating resolutions tested, details of coupling (with/without heat flux coupling), seeding scheme, duration of simulation, reference/s. This would clearly establish the novelty and leading position this the present model development and study.
- A new table has been added to the manuscript, giving a brief summary of coupled climate-iceberg models (Table 1).

- Introduction, from line 49. Provide more justification for focus on Antarctica and the Southern Ocean in a global model, given the potential importance of Greenland icebergs for dense water formation and properties in the North Atlantic.
- While the model allows for interactive icebergs in both hemispheres, our simulations only include iceberg in the Southern Ocean. We acknowledge the potential implications of iceberg-related freshwater and heat fluxes for deep-water formation in the North Atlantic and, hence, on AMOC. However, our main focus is an enhanced understanding of processes involved in climate iceberg interactions rather than simulating realistic climatologies. Some explanation is added at L62-65 + L330-334.

- 5, line 120: clarify what you mean by 'continuous', through time, or over space?
- Thank you for pointing out this ambiguity. It means continuous in space. The sentence has been clarified (L154).

- 5, lines 122-123, and in Fig. 1 caption: I do not quite follow 'integration of the calving flux over each basin to get total amount of ice discharge', and Fig. 1a; can this be further explained?
- The total discharge within one basin is calculated by integrating (summing up) discharge fluxes over all grid cells within this basin. The manuscript has been adapted accordingly (L157-158).

- 11, Fig. 7: There is considerable multidecadal-to-centennial variability in ocean temperature and AABW indices; while this is not a focus of the study, it is perhaps worthy of more comment and even some explanation.
- This variability is due to internal model variability. A brief statement is added to the manuscript (L262).

- 12, line 208: correct as 'shelf-region'
- Thanks for pointing this out. The manuscript has been corrected accordingly (L264).

- 13, line 219: do you mean 'pushed onshore due to Ekman dynamics'? (geostrophy sets up long-shore drift
- Yes, you are right. Thanks for pointing this out! The manuscript has been corrected accordingly (L280).

- 14, lines 250-251: Remarking that 'realistic representation of AABW formation along continental shelves is not feasible in out model setup', can this be further explained; it is a consequence of missing ice shelf cavities, or limited resolution, or both?
- The lack of a realistic representation of continental overflow is due to spurious mixing along steep topography gradients. Although we use a partly terrain following coordinate system, spurious mixing is strong and deep water formation occurs predominantly due to vertical mixing in the open ocean. This can also be seen in the Fig. A5 that shows the mixed layer depth. Deep water formation by open-ocean deep convection is a common bias in GCM's (Heuzé, 2021) and is linked to biases in potential density (compare Fig. A6). Both, the strong vertical mixing in the Weddell Sea and the strong bias in potential density, are reduced by including interactive icebergs. A brief statement about spurious mixing is added to L320f.

- 15, line 265: In discussing the delay of Southern Ocean greenhouse warming in climate projections due to cooling and freshening, do the authors imply the consequences of increasing iceberg calving? Perhaps be more explicit here.
- The latent heat flux due to iceberg melting is not necessarily included in common ESMs. Climate projections, including this feedback mechanism, hence might show delayed warming when compared to studies without iceberg-related heat fluxes. With increasing iceberg discharge, this effect can be expected to be further enhanced. A sentence has been added at L340f.

---

## Author Response (AR2)

General comments on the review:

I am very pleased by the work done during the review process. It clarifies well most the questions I had. I have minor comments on the review below.

We thank you for the constructive and positive comments.

Specific comments:

L116: I am unfamiliar with the 'lateral basal melt'. Could you reformulate or give more explanation? How it is written, it looks a different process to the buoyant convection melt term.

The lateral ‚basal' melt term accounts for the ‚basal' turbulent heat transfer that occurs on the submerged iceberg sides, motivated by Bigg et al. (1997). An explanation has been added to the manuscript (L108-111 in diff.pdf).

L181: by 'the nearest ocean grid cell and the neighbouring cells are added to this list', do you mean the nearest coastal ocean grid cell as suggested in l.132?

Icebergs are not seeded in coastal grid points. These grid cells are excluded from the list of potential iceberg seeding cells (like the saturated grid cells) to reduce the risk of instantaneous grounding. So, the term „neighbouring" means here, the neighbouring grid cells of a particular ocean grid cell that is nearest to the location of iceberg discharge, excluding coastal cells. An explanation has been added to the manuscript (L174f in diff.pdf).

L267: About the increased vertical mixing, I think it still need extra information. There is increase vertical mixing (figure on N2 / mixed layer) on the open ocean part of Amundsen Sea. I am not sure it is a direct effect of the presence of icebergs in the region as the iceberg melt will tend to stratify (and there is not many icebergs melt in the region). Based on your figure A8, I am wondering if the effect in Amundsen Sea is a consequence of the possible changes in gyre strength and extent or changes in front position.

The warming signal is most pronounced to a depth of around 100 m. To get a timeseries of this warm anomaly, we averaged ocean temperature over the upper 100 m between 110°W-130°W and 55°S-65°S. The timeseries shows a strong centennial variability but there is a persistent warm anomaly in ICB compared to CTL. The spatial distribution looks very similar to findings by Martin & Adcroft (2010) (s. Fig. 6b). However, there is no significant correlation between the temperature anomaly in this region and Weddell Gyre strength (r=-0.20), Drake Passage throughflow (r=-0.12), or the SAM index (r=-0.20).
A paragraph has been added to the manuscript (L313-319 in diff.pdf).

L307: I don't really understand how the feature to avoid saturation in ocean grid cell facilitated the fact that icebergs tend to stay at a single location. I naively thought that if it is more difficult for the icebergs to enter in a narrow bay where they get stuck, the icebergs will move more freely and so stay less in the Antarctic cold shelf seas.

Still icebergs may accumulate in certain areas, e.g. at the tip of the Antarctic Peninsula (Fig. 3). They may block the pathway for more downstream icebergs when a grid cell is saturated, although other model icebergs are not taken into account in one single iceberg's momentum balance. An explanation has been added to the manuscript (L300-302 in diff.pdf).

Figures and tables:

Table 1: the header of the last column is not clear (references -0.5ex>-0.5ex 0.5ex>0.5ex). What is 'ex'?

This issue is only present in the diff document and not in the actual manuscript. To be honest, I don't know what happened here. This error is not present in the revised version of the manuscript or the latest diff.pdf.

Figure 4: the bottom line is Sea Ice Height. What season it is?

Figure 4 shows the sea ice height averaged over the last 100 model years, no specific season. The caption has been changed for clarification (Caption Fig. 4).